# WIce-FOAM 1.0: Coupled dynamic and thermodynamic modelling of heterogeneous sea ice and waves using OpenFOAM-v2306

Rutger Marquart[1,2], Alberto Alberello[3], Alfred Bogaers[4,5], Francesca De Santi[6], and Marcello Vichi[1,2]

[1]Department of Oceanography, University of Cape Town, Cape Town, South Africa.
[2]Marine and Antarctic Research Centre for Innovation and Sustainability, Cape Town, South Africa.
[3]School of Engineering, Mathematics & Physics, University of East Anglia, Norwich, United Kingdom.
[4]Ex Mente Technologies, Pretoria, South Africa.
[5]Department of Mechanical and Aeronautical Engineering, University of Pretoria, Pretoria, South Africa.
[6]Institute of Applied Mathematics and Information Technologies of the National Research Council of Italy, Milan, Italy.

**Correspondence:** Rutger Marquart (rutger.marquart@uct.ac.za)

**Abstract.** We present WIce-FOAM 1.0, a numerical model built on OpenFOAM that couples the dynamics and thermodynamics of heterogeneous sea ice to analyse waves' response in marginal ice zone regions composed of consolidated ice floes and interstitial grease ice. The model represents prototypical conditions on the 5-kilometre scale, where each 10-metre grid cell classified as ice floe or grease ice may contain both ice types, but are predominantly occupied by one. Our model aims to study the mean shear viscosity of heterogeneous sea ice to bridge the gap with larger-scale ocean-sea ice models in which sub-grid details and wave effects are neglected. We tested the model in the Southern Ocean using a realistic sea-ice field from a SAR satellite image and complemented our analysis by idealised simulations. The thermodynamic model was coupled online to optimize the stiffness of the process scales and to explicitly account for the distinct characteristics of different ice types. We first investigated the dynamic response of sea ice to one-way wave forcing across a range of wave periods and directions. The results show that the domain-averaged sea-ice viscosity is scale invariant from approximately $800\,\mathrm{m}$ to $5\,\mathrm{km}$ and is primarily governed by the relative proportion of ice floes to grease ice, with less sensitivity to wave periods and directions. While the wave direction affects the local strain rate and viscosity, and the presence and orientation of narrow connections between the larger ice floes significantly influence the mean viscosity, these effects do not break the observed scale invariance. Finally, we demonstrate that, despite the different time scales, the mean viscosity responds nonlinearly to the inclusion of thermodynamic sea-ice growth. This model represents a first step towards a mechanistic understanding and description of heterogeneous sea ice, which is common in the Antarctic and is increasing in the warming Arctic. It can be used to design field experiments and to derive parametrisations of waves-in-ice response for large-scale sea-ice models.

## 1  Introduction

Antarctica has experienced relatively stable sea ice levels over recent decades and an apparent change of regime since 2017 (Diamond et al., 2024; Gilbert and Holmes, 2024; Wang et al., 2023). To understand the intricate response of Antarctic sea ice

to climate change (Golden et al., 2020), new-generation models should address variability in sea ice across smaller spatial and temporal scales (Iovino et al., 2022; Selivanova et al., 2024).

Sea ice is a dynamic heterogeneous medium, which is typically described as a mixture of ice constituents and open water areas (Barthélemy et al., 2016). However, away from the thicker pack ice regions, and especially in the Antarctic marginal ice zone, each type of ice has distinct material properties with pronounced thermal gradients (Tersigni et al., 2023).

During the winter season, the Antarctic sea ice cover expands over the Southern Ocean and reaches its maximum in September-October. In this phase, the ice constituents vary from newly formed ice in open water, which under the action of ocean waves transitions from a thin layer of frazil ice into a thicker, slushy layer known as grease ice, and eventually grows into more solid-like floes known as pancake ice floes (Wadhams et al., 2018; Nose et al., 2021). These pancake ice floes merge into aggregated ice floes and a more consolidated sea ice cover in the interior when the external forcing from ocean waves subsides.

During sustained warming conditions, ice-albedo feedback accelerates the melting of sea ice (Golden et al., 2020; Squire, 2020). Consequently, the heterogeneous sea ice cover weakens and disaggregates, and becomes more responsive to wind and ocean currents, also allowing for the propagation of waves deeper into the sea ice (Iovino et al., 2022; Squire, 2020; Thomson and Rogers, 2014). This creates a feedback loop, where larger ice floes break into smaller, mobile floes, increasing the transfer of momentum and energy into the sea ice (Thomson, 2022; Asplin et al., 2012). This process hinders the formation of consolidated sea ice, keeping the sea ice cover in a more heterogeneous condition (Day et al., 2024).

Most existing large-scale sea ice models adopt the continuum approach originally proposed by Hibler III (1979); Thorndike et al. (1975), which is generally considered valid over spatial scales of hundreds of kilometres (Mehlmann and Richter, 2017). Advancements in computing power in recent decades have enabled a shift toward higher-resolution models. This increased computational capacity has sparked renewed interest in exploring the properties of sea ice on spatial scales of tens of kilometres or less (Zhang, 2021), allowing us to apply and assess the continuum approach at finer resolutions.

Persistently challenging in continuum models is the necessity of integrating a suitable rheology for sea ice, i.e. the relationship between sea ice stress and deformation. Given the diverse spatial and temporal characteristics of the sea ice cover, there is no straightforward approach to accurately model sea ice dynamics using an effective large-scale rheology that accounts for all relevant processes (Åström et al., 2023). The need for introducing appropriate rheology for the various ice types has recently been highlighted (Herman, 2016; Golden et al., 2020; Skatulla et al., 2022). Different sea ice thicknesses and types experience specific growth and melting rates, i.e. thinner ice experiences faster growth and melting compared to thicker ice. Thinner ice is also more susceptible to mechanical deformation (Golden et al., 2020). Moreover, the presence and properties of frazil and grease ice and the concurrent action of waves are barely addressed (Dumont, 2022), largely due to limited research on frazil ice in terms of field observations (Paul et al., 2021).

Small-scale models are useful for informing the development of parametrisations for large-scale sea ice models. Marquart et al. (2021, 2023) introduced a two-dimensional small-scale computational fluid dynamics (CFD) model implemented in OpenFOAM, designed to represent heterogeneous sea ice conditions on the metre scale. The model distinguishes between two distinct ice types: ice floes and interstitial grease ice, as shown in Fig. 1(a), each characterised by unique material properties,

and captures sea ice dynamics under wavy conditions. Thermodynamic effects were excluded, as the simulations focused on short time scales of less than a minute.

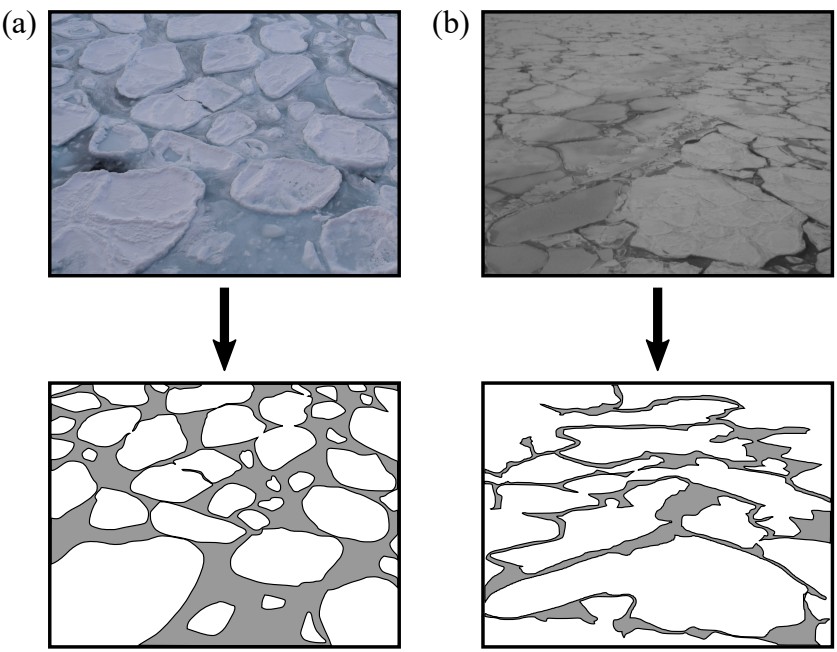

**Figure 1.** Comparison of the heterogeneous sea ice cover at two spatial scales: (a) the metre scale $(10-100\,\mathrm{m}^2)$, based on a photograph taken during the winter cruise in July 2017, where individual ice floes and grease ice are clearly distinguishable (Marquart et al., 2021, 2023), and (b) the kilometre scale $(1-10\,\mathrm{km}^2)$, based on a photograph taken during the SCALE winter cruise in July 2022, where regions identified as either ice floes or grease ice primarily consist of one type, though both may be present.

Here, we extend the model developed by Marquart et al. (2021, 2023) by expanding the spatial domain to the kilometre scale. At this scale, individual ice floes and grease ice patches can no longer be explicitly resolved, as illustrated in Fig. 1(b). Nonetheless, representing the heterogeneous nature of the sea ice cover remains essential. To accommodate this, we reinterpret the definitions of 'ice floe' and 'grease ice' within the model: a region is classified as 'ice floe' if it predominantly consists of ice floes, even if grease ice is present. Conversely, the same applies to grease ice.

Additionally, at this kilometre scale, we incorporate thermodynamic processes, which require extending the time window to the order of days. The aim is to capture the evolution of key sea ice variables, such as thickness and viscosity, across space and time, which are particularly relevant for upscaling to large-scale, global sea ice models.

The remainder of this work is structured as follows. Section 2.1 describes the dynamic model and its implementation within the OpenFOAM framework. Section 2.2 introduces the thermodynamic model, based on the formulation by Tedesco et al. (2009) and the frazil ice production equation from Haarpaintner et al. (2001). The coupling approach between the dynamics and thermodynamics models is presented in Section 2.3. Section 2.4 outlines the model configuration used in the simulations,

and results are discussed in Section 3. A detailed discussion of the key findings is provided in Section 4. Finally, Section 5 summarises the work.

## 2 Methodology

### 2.1 Dynamic OpenFOAM model

OpenFOAM is an open-source computational fluid dynamics (CFD) software toolbox used to solve a wide range of problems, including complex fluid flows, heat transfer and solid mechanics (Medina et al., 2015). The software is highly customizable and extensively used in both research and industry.

Our proposed two-dimensional continuum framework, WIce-FOAM 1.0, is implemented using OpenFOAM-v2306 and formulated in the $(x, y)$-plane using the finite volume method (FVM) and the volume of fluid (VOF) method. The FVM discretizes

the domain into a finite number of control volumes, or cells, and applies conservation laws to each cell (Ferziger et al., 2019). The VOF method is a numerical FVM technique to describe the interface between two immiscible and incompressible materials. In our model, we use the IsoAdvector method as the VOF implementation (Roenby et al., 2016). Both methods are key to the model and are important for coupling the dynamics and thermodynamics models in Section 2.3.

Within this framework, we consider a domain with $100\%$ sea ice concentration and investigate the impact of sea ice hetero-

85 geneity on both the dynamic response to wave forcing (in a one-way coupled setup) and on thermodynamic processes. Phase transitions between different ice types are not included at this stage. We model a heterogeneous sea ice cover composed of ice floes and grease ice. Each cell is classified as either 'ice floe' or 'grease ice' according to the dominant ice type within that cell, determined from the initial condition field derived from a synthetic aperture radar (SAR) image (Section 2.4), even though both types may be present within a single cell in the real system.

Grease ice is treated as a highly viscous fluid that dissipates energy during ice floe collisions. As a result, interactions between ice floes are represented as continuous, churning contact of varying intensity rather than brief, forceful impacts. This justifies the exclusion of ice floe failure and fracture due to floe-floe collisions. Other potential fracture mechanisms are also excluded, such as those driven by ice convergence, shear deformation, or wave-induced bending. Lastly, we focus our analysis on one-way wave-ice interactions: a harmonic propagating wave forcing is imposed on the sea ice domain, which excludes

wave dissipation effects.

#### 2.1.1 Momentum balance equation

The momentum transport of incompressible sea ice is formulated in the momentum balance equation as

$$\rho \left( \frac{\partial h \boldsymbol{U}}{\partial t} + (\boldsymbol{U} \cdot \nabla) h \boldsymbol{U} \right) = \boldsymbol{\tau}_w + \boldsymbol{\tau}_a + \nabla \cdot \boldsymbol{\sigma}, \tag{1}$$

where $\boldsymbol{U}$ represents the two-dimensional sea ice velocity vector, $h$ is the sea ice thickness, and $t$ is time. External forces, in the

100 form of the in-plane oceanic wave and wind stresses, act on the sea ice cover and are denoted as $\boldsymbol{\tau}_w$ and $\boldsymbol{\tau}_a$, respectively. The Cauchy stress tensor, $\boldsymbol{\sigma}$, characterizes the stress state of the sea ice and differs by ice type, denoted $\boldsymbol{\sigma}_f$ for ice floes and $\boldsymbol{\sigma}_g$ for

grease ice.

The sea ice mass per unit area, $m$, is

$$m = \rho h, \tag{2}$$

where $\rho$ denotes the sea ice density. The evolution of sea ice thickness (Hibler III, 1979; Hutchings et al., 2004), $h$, is given as

$$\frac{\partial h}{\partial t} + \nabla \cdot (h\boldsymbol{U}) = 0, \tag{3}$$

which extends the model proposed by Marquart et al. (2021, 2023), wherein a constant sea ice thickness was assumed.

The external in-plane oceanic wave stress, denoted as $\boldsymbol{\tau}_w$, is derived from the linear wave theory (Holthuijsen, 2010; Herman, 2018). This stress is characterized by the presence of two distinct components

$$\boldsymbol{\tau}_w = \boldsymbol{\tau}_{sd} + \boldsymbol{\tau}_{fk}, \tag{4}$$

where the skin drag, $\boldsymbol{\tau}_{sd}$, represents the viscous effects across the ice-ocean interface, and the Froude-Krylov force, $\boldsymbol{\tau}_{fk}$, results from the wave pressure field acting on the submerged surface of the ice floes.

The quadratic skin drag (Hutchings, 2000) is given as

$$\boldsymbol{\tau}_{sd} = \rho_w C_w |\boldsymbol{U}_w - \boldsymbol{U}| ((\boldsymbol{U}_w - \boldsymbol{U})\cos\theta_w + (\boldsymbol{U}_w - \boldsymbol{U}) \times \boldsymbol{k}\sin\theta_w), \tag{5}$$

where $\rho_w$ represents the water density, and $\theta_w$ is the ice-ocean turning angle. The unit normal vector to the ice surface is denoted by $\boldsymbol{k}$, and $C_w$ is the ice-ocean drag coefficient, which varies between the two ice types. The orbital wave velocity of the water, $\boldsymbol{U}_w$, for monochromatic waves is defined as

$$\boldsymbol{U}_w = \begin{pmatrix} U_{w_x} \\ U_{w_y} \\ U_{w_z} \end{pmatrix} = \begin{pmatrix} a\omega\sin(\omega t - k_x x) \\ a\omega\sin(\omega t - k_y y) \\ a\omega\cos(\omega t - k_x x - k_y y) \end{pmatrix}, \tag{6}$$

where $x$, $y$ and $z$ represent the Cartesian coordinates within the sea ice domain, with $z$ denoting the vertical direction. The model is two-dimensional, the vertical component of the wave velocity does not enter the governing momentum balance equation, but is included here for completeness. The parameters $a$, $\omega$, and $k$ correspond to the wave amplitude, wave frequency, and wave number, respectively. The wave frequency, $\omega$, and wave number, $k$, are given by $\omega = 2\pi/T$ and $k = 2\pi/\lambda$, derived from the wave period, $T$, and wavelength, $\lambda$, using the deep water dispersion relation, $\omega^2 = gk$, where $g$ denotes the gravitational acceleration. The wave numbers in $x$-, and $y$-direction are formulated as

$$k_x = k\cos(\theta_{wa}), \qquad\qquad\qquad k_y = k\sin(\theta_{wa}), \tag{7}$$

where $\theta_{wa}$ denotes the wave direction angle, measured relative to the $x$-axis and defined as positive in the counter-clockwise direction. A value of $\theta_{wa} = 0°$ corresponds to wave propagation exclusively along the $x$-axis.

The in-plane form drag, acting on the ice floe circumference due to velocity differences between the floes and surrounding grease ice, is implicitly captured by the continuum approach, which includes both ice constituents and enforces velocity continuity at the interface.

 The Froude-Krylov force, $\boldsymbol{\tau}_{fk}$, acts in the basal plane (the lower ice surface in contact with the ocean parallel to the $xy$-plane) and represents the horizontal surge force generated by the wave-induced pressure (Herman, 2018) at the interface between the ice floe and the water. This is:

$$\boldsymbol{\tau}_{fk} = -\int_{h_w} p\boldsymbol{n}\,\mathrm{d}z, \tag{8}$$

where $h_w$ is the height of the submerged portion of the ice floe thickness, and $\boldsymbol{n}$ is the outward pointing unit vector, normal to
 the ice floe circumference. The wave-induced pressure, denoted as $p$, is

$$p = \rho_w g a \sin(\omega t - kx). \tag{9}$$

The atmospheric forcing is represented by a wind stress (Hutchings, 2000), $\boldsymbol{\tau}_a$, applied to the upper (apical) surface of the ice exposed to the atmosphere (also parallel to the $xy$-plane), and is expressed as

$$\boldsymbol{\tau}_a = \rho_a C_a |\boldsymbol{U}_a|(\boldsymbol{U}_a \cos\theta_a + \boldsymbol{k} \times \boldsymbol{U}_a \sin\theta_a), \tag{10}$$

 where $\boldsymbol{U}_a$ is the wind velocity, $\rho_a$ denotes the air density, $C_a$ is the ice-air drag coefficient, and $\theta_a$ is the wind turning angle. Note that this term is analogous to the ocean drag formulation. However, since sea ice drift velocities are typically much smaller than wind speeds, the difference is generally negligible (Leppäranta, 2011).

### 2.1.2 Sea ice rheology

The sea ice rheology in the model defines the relationship between internal ice stresses, denoted by $\boldsymbol{\sigma}$, and ice deformation,
 expressed in terms of the strain rate, $\dot{\boldsymbol{\epsilon}}$. In accordance with the infinitesimal small strain theory, the strain rate tensor can be expressed in relation to the sea ice velocity gradient, $\boldsymbol{\nabla U}$, as

$$\dot{\boldsymbol{\epsilon}} = \frac{1}{2}(\boldsymbol{\nabla U} + (\boldsymbol{\nabla U})^T). \tag{11}$$

Each component is characterized by its own rheology, with ice floes displaying solid-like behaviour and grease ice exhibiting fluid-like behaviour. Marquart et al. (2021, 2023) used a 'Hookean-like' flow rule to describe the constitutive law for the
 solid-like behaviour of ice floes. While the model effectively reproduced solid-like behaviour, it did not account for elastic unloading.

In this study, we replace the 'Hookean-like' ice floe rheology with one that incorporates viscous behaviour. Therefore, the ice floes follow the viscous-plastic rheology proposed by Hibler III (1979):

$$\boldsymbol{\sigma}_f = 2\eta_f\dot{\boldsymbol{\epsilon}} + \boldsymbol{I}\left((\zeta_f - \eta_f)\mathrm{tr}(\dot{\boldsymbol{\epsilon}}) - \frac{P_f}{2}\right), \tag{12}$$

where $\boldsymbol{\sigma}_f$ denotes the Cauchy stress tensor for ice floes. The shear and bulk viscosities of ice floes are represented by $\eta_f$ and $\zeta_f$, respectively. The internal ice floe strength is $P_f$ and $\boldsymbol{I}$ represents the identity tensor. The two strain rate-dependent viscosities are coupled via

$$\zeta_f = \frac{P_f}{2\Delta}, \qquad\qquad\qquad \eta_f = \frac{\zeta_f}{e^2}, \qquad\qquad (13)$$

where $e$ indicates the eccentricity, the ratio between the in-plane principal axes of the elliptical yield curve (Hibler III, 1979). The effective strain rate, $\Delta$, is

$$\Delta = \sqrt{(\dot{\epsilon}_{11}^2 + \dot{\epsilon}_{22}^2)(1 + e^{-2}) + 4e^{-2}\dot{\epsilon}_{12}^2 + 2\dot{\epsilon}_{11}\dot{\epsilon}_{22}(1 - e^{-2})}, \qquad\qquad (14)$$

where $\dot{\epsilon}_{11}$, $\dot{\epsilon}_{22}$ and $\dot{\epsilon}_{12}$ denote the Cartesian components of the symmetric strain rate tensor. As the strain rate approaches zero, the viscosity becomes unbounded. To prevent this singularity, we impose a lower bound on the effective strain rate, $\Delta_{min} = 2 \cdot 10^{-6}$ s$^{-1}$. The modified internal ice floe strength, $P_f$, can be written as

$$P_f = P_f^* h, \qquad\qquad (15)$$

where $P_f^*$ represents an empirical constant. Equation (15) differs from Hibler III (1979) by excluding the compactness parameter, which is unnecessary in our model due to the explicit treatment of the two ice phases.

In Marquart et al. (2021, 2023) grease ice was governed by a viscous-plastic (VP) material law, which is similar to the rheology developed by Hibler III (1979); Thorndike et al. (1975). However, Marquart et al. (2021, 2023) observed a singularity in the viscosity and strain rate of grease ice associated with the passage of waves. This singularity is characterized by locally very high viscosity values, linked to strain rate values approaching zero. To address unnatural behaviour due to this singularity, we present here a revised rheology for the grease ice.

In this study, grease ice is assumed to behave as an incompressible fluid, based on literature indicating that it is primarily composed of water (Smedsrud, 2011; Mackie et al., 2020). Consequently, its rheology is represented as that of an incompressible, non-Newtonian viscous fluid (Newyear and Martin, 1997; Eberhard et al., 2019):

$$\boldsymbol{\sigma}_g = 2\eta_g \dot{\boldsymbol{\epsilon}}, \qquad\qquad (16)$$

where the Cauchy stress tensor for grease ice is indicated as $\boldsymbol{\sigma}_g$. The shear viscosity, $\eta_g$, follows the Cross model incorporating shear thinning, wherein viscosity is constrained for extremely high and low strain rate values (Hauswirth et al., 2020; Eberhard et al., 2019; Galindo-Rosales et al., 2010). The shear viscosity, $\eta_g$, is

$$\eta_g = \eta_\infty + \frac{\eta_0 - \eta_\infty}{1 + (C|\dot{\gamma}|)^m}, \qquad\qquad (17)$$

where $C$ represents the Cross time constant, $|\dot{\gamma}| = \sqrt{2\dot{\boldsymbol{\epsilon}} : \dot{\boldsymbol{\epsilon}}}$ the shear rate magnitude, and $m$ the degree of shear thinning. Inertia is likely to contribute to the effective viscosity of grease ice (de Carolis et al., 2005). Therefore, we assume that $\eta_\infty$ and $\eta_0$ are

thickness-dependent infinite and zero shear viscosities of grease ice, and follow a power law:

$$\eta_0 = \eta_{0_c} \left( \frac{h}{h_{g_0}} \right)^{\alpha_0}, \qquad\qquad \eta_\infty = \eta_{\infty_c} \left( \frac{h}{h_{g_0}} \right)^{\alpha_\infty}, \qquad\qquad (18)$$

where $\eta_{\infty_c}$ and $\eta_{0_c}$ are the reference infinite and zero shear viscosities of grease ice, respectively, at the reference thickness of grease ice, $h_{g_0}$. The power law exponents $\alpha_\infty$ and $\alpha_0$ describe how $\eta_\infty$ and $\eta_0$ change with thickness.

Parameter values associated with the dynamic model are summarized in Table 1, which is organised into three groups, namely
ice floe rheology parameters, grease ice rheology parameters, and wave-related parameters. Ice floe rheology parameters are based on the model by Hibler III (1979) and related formulations (e.g. Mehlmann and Richter, 2017). Notably, the limit of the effective strain rate (Leppäranta and Hibler III, 1985) is deliberately reduced by one order of magnitude to ensure numerical stability. Although this adjustment affects the solidity of the ice floes by modifying their viscosity, the overall simulation results remain largely unaffected. Values for the grease ice rheology are derived from literature sources such as Paul et al. (2021), and
further refined through empirical tuning via iterative simulations. Wave-related parameters include the ice-ocean turning angle, set to zero to reflect that the water drag on the ice acts purely along the flow direction. Drag coefficient values associated with wave characteristics are taken from Smedsrud (2011) and Alberello et al. (2020). Finally, a range of wave parameters is selected to conduct a sensitivity analysis in Section 3.

**Table 1.** Parameters related to the dynamic model.

| Parameter | Definition | Value | Unit |
|---|---|---|---|
| $P_f^*$ | ice floe strength parameter | 27500 | $\mathrm{N\,m^{-2}}$ |
| $e$ | eccentricity | 2 | - |
| $\Delta_{min}$ | limit of the effective strain rate | $2 \cdot 10^{-6}$ | $\mathrm{s^{-1}}$ |
| $\eta_{0,\infty_c}$ | reference viscosities for grease ice | 600, 6 | $\mathrm{kg\,s^{-1}}$ |
| $h_{g_0}$ | reference thickness of grease ice | 0.1 | m |
| $\alpha_{0,\infty}$ | zero, infinite shear exponent | 0.5 | - |
| $C$ | Cross time constant | 50 | s |
| $m$ | degree of shear thinning | 0.5 | - |
| $\rho_{w,f,g}$ | density sea water, ice floes, grease ice | 1026, 909, 916 | $\mathrm{kg\,m^{-3}}$ |
| $C_{w_{f,g}}$ | ice-ocean drag coefficient ice floes, grease ice | 0.005, 0.006 | - |
| $\theta_w$ | ice-ocean turning angle | 0 | $^\circ$ |
| $T$ | wave periods | 8.8, 12.4, 15.2 | s |
| $a$ | wave amplitude | 0.8 | m |
| $\lambda$ | wavelengths | 120, 240, 360 | m |
| $\theta_{wa}$ | wave direction angles | 0, 90, 180, 270 | deg |

## 2.2 Thermodynamic model

As with the rheologies of ice floes and grease ice, discussed in Section 2.1.2, this study applies two distinct thermodynamic models to the two ice types.

For the ice floes, the one-dimensional thermodynamic model in the $z$-direction, developed by Tedesco et al. (2009), is applied to OpenFOAM cells associated with ice floes to simulate thermodynamic variations in snow and ice thickness. The model accounts for multiple layers of sea ice (including columnar ice, snow ice, and superimposed ice) and/or snow, as well as surface heat fluxes. These layers are assumed to be in thermal equilibrium, with interface temperatures determined by the continuity of heat fluxes:

$$\rho_i c_i \frac{\partial T_i}{\partial t} = \frac{\partial}{\partial z}\left(K_i \frac{\partial T_i}{\partial z}\right) - \frac{\partial}{\partial z}[I_{pen}(z)], \tag{19}$$

where $\rho$ is the density, $c$ the specific heat and $T$ the temperature of the layers (the subscript $i$ denotes either snow or the sea ice layers). The thermal conductivity is defined by $K$, and $I_{pen}$ is the flux of penetrating solar radiation through each layer. The vertical coordinate is defined as positive in the downward direction, where $z = 0$ corresponds to the top surface. The penetrating radiation can be written as

$$I_{pen}(z) = I_0 \exp(-\kappa_i z), \tag{20}$$

where $I_0$ indicates the penetrating solar flux at the top ice/snow surface and $\kappa_i$ the extinction coefficient. The new temperatures for each layer are calculated using a finite-difference formulation of Eq. (19), with the calculation being performed through an iterative process. Growth and melting rates are determined from expressions for the enthalpy of snow and sea ice. The enthalpy of snow (or fresh ice) per unit volume (Hunke et al., 2015), $q_s$, is

$$q_s(T) = -\rho_s(-c_0 T + L_0), \tag{21}$$

where $\rho_s$ is the snow density, $c_0$ the specific heat of fresh ice, and $L_0$ the latent heat of fusion of fresh columnar ice. The enthalpy of sea ice is not straightforward due to the presence of brine pockets, where salinity varies inversely with temperature. Assuming a predetermined value for salinity, a direct relationship between temperature and the enthalpy of sea ice per unit volume can be obtained:

$$q_i(T) = -\rho_i \left( c_0(T_m - T) + L_0\left(1 - \frac{T_m}{T}\right) - c_w T_m \right), \tag{22}$$

where $\rho_i$ is the sea ice density, $T_m$ the temperature at which the ice is completely melted, and $c_w$ the specific heat of seawater. Once the enthalpy is obtained, the surface growth/melting is:

$$q_{s,i}\Delta h_{s,i} = \begin{cases} (F_0 - F_{ct})\Delta t & \text{if} \quad F_0 > F_{ct} \\ 0 & \text{otherwise} \end{cases}, \tag{23}$$

where $\Delta h_{s,i}$ represents the thickness change in snow or sea ice, respectively, over the time step $\Delta t$. The net surface heat flux from the atmosphere to the ice, $F_0$, represents the resultant of all fluxes included in the thermodynamic model (Hunke et al.,

2015). These comprise the sensible heat flux, latent heat flux, incoming and outgoing long wave radiation, and incoming short wave radiation. $F_{ct}$ denotes the conductive flux from the top surface to the interior of the ice.

The growth and melting occurring at the bottom layer of ice is

$$q_i \Delta h_i = (F_{cb} - F_{bot}) \Delta t, \tag{24}$$

where $F_{cb}$ is the conductive heat flux at the bottom surface and $F_{bot}$ is the net downward heat flux from the ice to the ocean.

For the cells in OpenFOAM associated with grease ice, a different thermodynamic approach is applied. We implemented 240 the frazil ice production rate proposed by Haarpaintner et al. (2001) in the thermodynamic model (Tedesco et al., 2009):

$$\Delta h_f = \frac{F_0}{\rho_f L_f} \Delta t, \tag{25}$$

where the rate of change of frazil ice thickness depends on the net surface heat flux, $F_0$, a constant value of frazil ice density, $\rho_f$, and latent heat of fusion specific for frazil ice, $L_f$.

In the thermodynamic model (Tedesco et al., 2009), most variables, including changes in snow and sea ice thickness, depend on the state at the previous time step, thereby necessitating the tracking of sea ice variables across time steps. This temporal tracking is particularly important for the coupling approach discussed in Section 2.3. The model is forced with atmospheric input variables for the selected location, including total cloud cover fraction, specific humidity of air and surface, air temperature, wind velocity components, downward surface solar radiation flux and the mean total precipitation rate, all sourced from 250 ERA5 (Hersbach et al., 2018a, b), although alternative datasets can also be used.

Thickness evolution results from the thermodynamic model, based on the formulations by Tedesco et al. (2009), are presented in Fig. 2(a), which depicts a complete seasonal cycle within 2022. The evolution of snow and sea ice thickness are shown in blue and red, respectively. Figure 2(b) provides a detailed view of July, comparing the evolution of sea ice thickness with frazil 255 ice thickness, derived from Haarpaintner et al. (2001). Fig. 2 shows that the thickness of both sea ice and snow is zero, i.e. no ice, during the summer months as the air temperature exceeds the threshold for ice formation and growth. Ice begins to form and its thickness to increase in winter (June) when the air temperature drops. Note that in the model, snow growth commences only once the minimum snow threshold ($h_{s,\min} = 0.02 \, \text{m}$) has been exceeded, see Fig. 2(b). Sea ice thickness returns to zero in spring, after the sea ice melting. As expected, frazil ice exhibits a higher growth rate than sea ice. This is primarily because the 260 net surface heat flux, $F_0$, increases for thinner ice, accelerating the rate of thickness change. Additionally, the lower latent heat of fusion of frazil ice compared to ice floes further enhances its growth. We observe that the thickness of frazil ice exceeds that of sea ice. While this outcome is physically unrealistic due to the expected phase transition, it does not affect our study, as our simulation results in Section 3 will be limited to a one-day period rather than an entire seasonal cycle.

The main parameter values associated with the thermodynamic model are summarized in Table 2. Only the constant parameters 265 in the equations presented in this work are included; all other equations, parameters, as well as initial conditions, are available in the supplementary code.

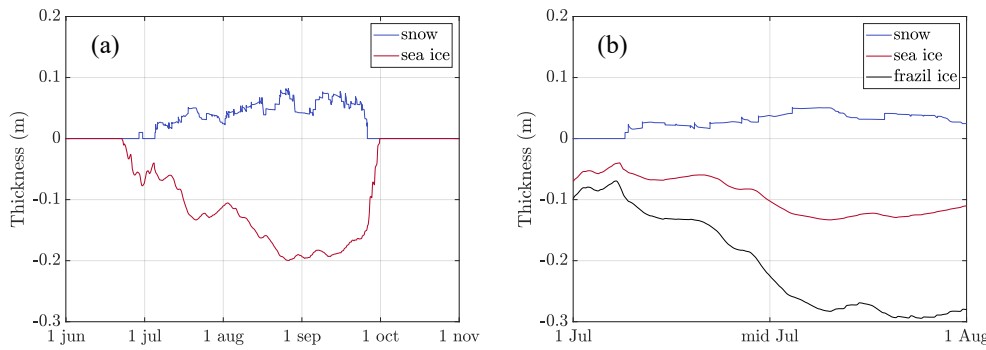

**Figure 2.** Results from the thermodynamic model, illustrating (a) the thickness evolution from June to November 2022, and (b) a comparison of snow and sea ice thickness evolution with frazil ice development in July 2022.

**Table 2.** Parameters related to the thermodynamic model.

| Parameter | Definition | Value | Unit |
|:---:|:---|:---:|:---:|
| $c_0$ | specific heat of fresh ice | 2093 | $\mathrm{J\,(kg\,K)}^{-1}$ |
| $c_w$ | specific heat of seawater | 4186 | $\mathrm{J\,(kg\,K)}^{-1}$ |
| $L_0$ | latent heat of fusion of fresh ice | 297000 | $\mathrm{J\,kg}^{-1}$ |
| $L_f$ | latent heat of fusion of frazil ice | 234000 | $\mathrm{J\,kg}^{-1}$ |
| $T_m$ | melting temperature of ice | 273.15 | K |
| $\Delta t$ | time step | 450 | s |

## 2.3 Coupling between the dynamics and thermodynamics models

Thermodynamic contributions can be seamlessly integrated into the dynamic model within OpenFOAM by manipulating the thickness variable and utilizing the VOF approach in the dynamic model to explicitly differentiate between ice types. The coupling between dynamics and thermodynamics is achieved by alternately running the dynamics and thermodynamics models within a for-loop.

The main challenge in the coupling arises from integrating both models that operate on distinct temporal scales, for which conventional up- or down-sampling methods to achieve a common timescale are considered impractical. The dynamic model represents a fast process, capturing the interaction between sea ice and a harmonically propagating wave. To accurately resolve the wave characteristics, the dynamic model requires a time step shorter than the wave period, typically in the order of seconds. By contrast, the thermodynamic model evolves on a much slower timescale, as processes of sea-ice melting and growth occur gradually over hours rather than seconds, and a time step of a few minutes is sufficient. To further improve computational efficiency, a well-established approach based on thickness categories is adopted (Sun and Solomon, 2024). In this method, sea ice is divided into a discrete set of thickness ranges, each representing ice of similar physical properties. The cells within the $xy$-plane are then grouped accordingly, and the average thickness for each category is computed. These category-averaged

values are then used as inputs to the thermodynamic model, which is executed once for each thickness range, plus an additional run for grease ice, thereby reducing the number of required model runs.

We note that the motion of sea ice subjected to a harmonically propagating wave exhibits periodic behaviour (Marquart et al., 2023). Therefore, we can assume that the dynamic response becomes periodic after one full wave period. Figure 3 illustrates a schematic of the coupling approach, showcasing the first two iterations in the for-loop. The simulation begins with a spin-up phase involving only the dynamics, during which the system evolves toward equilibrium under periodic wave forcing. Equilibrium for the dynamic component is considered reached when oscillations in sea ice velocity repeat over one wave period without any net change in velocity. Typically, the required spin-up time corresponds to approximately four to five wave periods (denoted $t_n$). After the spin-up phase, coupling between dynamics and thermodynamics begins at $t_n$, initiated by a for-loop, with the two models being executed alternately.

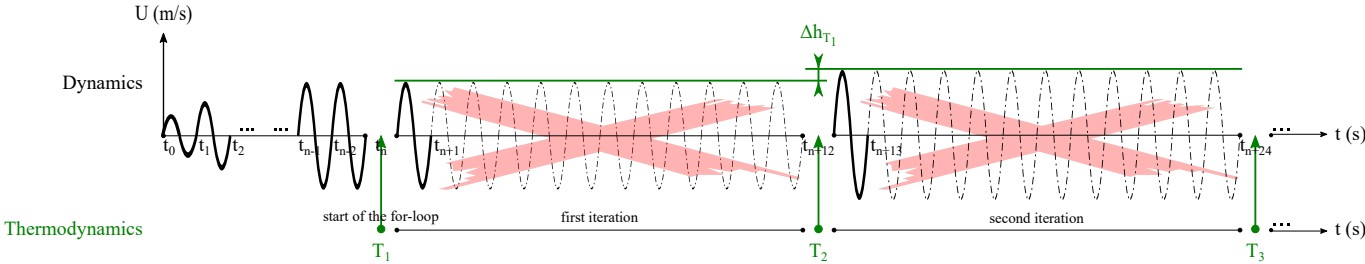

**Figure 3.** Schematic of the coupling between the dynamics and thermodynamics models illustrating the first two iterations in the for-loop.

The first iteration begins with the execution of the thermodynamic model over one thermodynamic interval, defined as the time between $T_1$ and $T_2$. The resulting thickness change, $\Delta h_{T_1}$, is initially stored. Subsequently, the dynamic model resumes from the end of the spin-up phase and runs for one dynamic interval - equivalent to a single wave period (indicated in boldface in Fig. 3) - as successive, identical wave periods within the thermodynamic interval are not simulated (represented by dashed oscillations and marked with a thick cross in Fig. 3). The thickness in the dynamic model is then updated using the stored output from the thermodynamic model (indicated by $\Delta h_{T_1}$ in Fig. 3), resulting in a change in sea ice velocity. This concludes the first iteration. The second iteration begins with the thermodynamic model advancing over the next thermodynamic interval, from $T_2$ to $T_3$, and the sequence repeats until the for-loop is completed.

One of the objectives of the present work is to demonstrate that coupling between the dynamics and thermodynamics models is necessary. This is assessed by evaluating the linearity of their relationship with respect to sea ice thickness and sea ice viscosity using the following equations:

$$\underbrace{\bar{h}_D + \sum \Delta h_T}_{decoupled} = \underbrace{\bar{h}_C}_{coupled} \, , \qquad\qquad \underbrace{\bar{\eta}_D + \sum \Delta \eta_T}_{decoupled} = \underbrace{\bar{\eta}_C}_{coupled} \, . \qquad (26)$$

If the dynamics and thermodynamics act independently in a decoupled manner, then the domain-averaged sea ice thickness and viscosity derived from the dynamic model, $\bar{h}_D$ and $\bar{\eta}_D$, combined with the cumulative thermodynamic changes per time

step, $\Delta h_T$ and $\Delta \eta_T$, should be equivalent to the domain-averaged thickness and viscosity of the coupled model, $\bar{h}_C$ and $\bar{\eta}_C$. Any deviation between the left- and right-hand sides of Eq. (26) would indicate nonlinear behaviour in the evolution of these variables, highlighting the significance of the coupling.

The thermodynamic changes per time step, $\Delta h_T$ and $\Delta \eta_T$, can be obtained by running the thermodynamic model separately. However, we employ the coupled model with the wave amplitude set to zero for simplicity. This ensures that sea ice dynamics are excluded, while preserving the correct category proportions across the domain.

## 2.4 Model configurations

We design and test a few configurations to showcase the dynamics of the ice floe-grease ice heterogeneous system and to demonstrate the effects of the coupling.

To realistically distribute the sea ice types within the computational domain, we derive the full-field configuration from a synthetic aperture radar (SAR) image (Fig. 4(a)) acquired by the COSMO-SkyMed (CSK) satellite on July 22, 2022, at 07:02 UTC, in support of the SCALE-WIN22 research expedition (Vichi, 2023). The SAR image provides the intensity of the reflected radar signal, which can be assimilated to derive surface properties. The open ocean and ice-covered regions are clearly distinguishable, with the open ocean appearing darker. For our analysis, we select a subregion within the ice-covered area, outlined by the green rectangle in the figure. A comparison with the sea ice concentration derived from AMSR2 satellite on the same day (Fig. 4(b)) confirms that this subregion corresponds to an area of $100\%$ sea ice concentration at the 25-kilometre scale. In the region of interest (Fig. 4(c)), distinct wave patterns are clearly visible that may confound the retrieval of heterogeneity. To remove these wave signatures, a mask is applied in Fourier space. The resulting intensity variations are then interpreted as differences in ice type. Pixels with a filtered amplitude greater than the median are classified as ice floes, while those with lower amplitude are identified as interstitial grease ice. We observed that using a different threshold would change the relative distribution of ice floes and grease ice. Therefore, we tested this by comparing model results across different subregions of the domain, each containing varying proportions. Figure 4(d) shows the upper-left corner of the selected region following the binarisation process. The final result is a $504 \times 504$ grid with a 10-metre pixel resolution.

Thickness information cannot be obtained from Fig. 4, therefore, reference visual observations collected during the SCALE-WIN22 research expedition (Vichi, 2023) were used to supplement the analysis.

Figure 5(a) shows the initial thickness prescribed at $t = 0\,\mathrm{s}$ for the full-field case. The sea ice in the region of the SAR image had a variable thickness ranging from a few centimetres for grease ice to $0.4-0.5\,\mathrm{m}$ for pancake ice. Since most sea-ice models do not simulate ice formation starting from frazil ice aggregation (Smedsrud, 2011; Smedsrud and Martin, 2015), a minimum thickness threshold of $0.1\,\mathrm{m}$ is typically used (e.g. Rousset et al., 2015). Accordingly, we randomly initialised three ice floe thicknesses just above this threshold, $h_f = 0.15\,\mathrm{m}$, $h_f = 0.13\,\mathrm{m}$, and $h_f = 0.11\,\mathrm{m}$, along with a grease ice thickness slightly below the smallest ice floe thickness, set to $h_g = 0.08\,\mathrm{m}$. The use of thinner ice is also more compatible with the simplified experimental design in which waves' propagation is not affected by the ice medium. The domain, measuring $5040 \times 5040\,\mathrm{m}^2$, is discretised using a uniform grid with cell dimensions of $10 \times 10\,\mathrm{m}^2$.

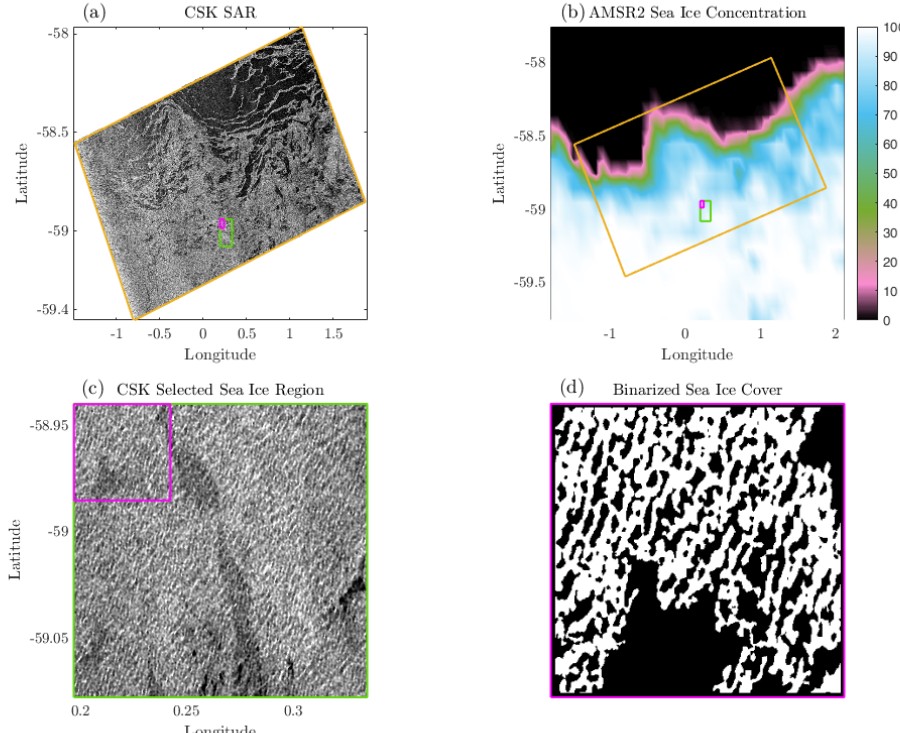

**Figure 4.** Sea ice distribution derived from a SAR image, illustrating (a) SAR image acquired by the COSMO-SkyMed (CSK) satellite on July 22, 2022, at 07:02 UTC, (b) sea ice concentration from AMSR2 on the same day, (c) selected $1024 \times 1024$ pixel region corresponding to $100\%$ sea ice concentration, and (d) binarised sea ice domain used to initialize the full-field configuration.

The complementary test cases, shown in Fig. 5(b-e), represent an idealised version of Fig. 5(a), featuring large ice floes with narrow connections, a characteristic frequently observed in the full-field case. They are designed to clarify the behaviour observed in the full-field case in a more controlled and simplified setting. Two circular floes of different sizes and thicknesses are linked by a narrow connection, with thickness spatially varying from one floe to the other. In this configuration, we investigate the effect of the narrow connection and analyse the domain-averaged viscosity by varying its orientation with respect to the imposed wave (always from the west). The geometry scales were chosen in relation to the wave characteristics (see Table 1). The wiggles were included in Fig. 5(e) to test whether the irregular shape induces a significant difference in the domain-averaged viscosity results. The circular ice floes have a radii of $r = 60\,\mathrm{m}$ and $r = 120\,\mathrm{m}$, each with initial thicknesses of $h_f = 0.11\,\mathrm{m}$ and $h_f = 0.15\,\mathrm{m}$, respectively. The grease ice has the same thickness as in the full-field configuration. The domain dimensions are $720 \times 720\,\mathrm{m}^2$, discretised using a uniform grid with cell sizes of $2 \times 2\,\mathrm{m}^2$.

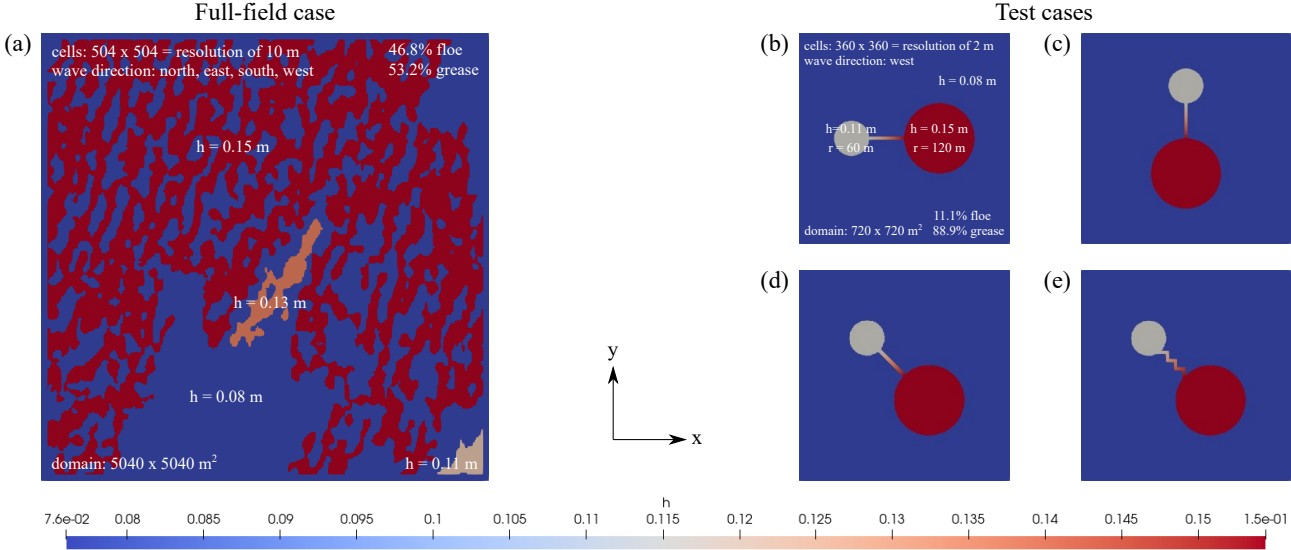

**Figure 5.** Initial layout along with additional information for (a) the full-field case, and (b-e) the test cases.

Periodic boundary conditions are applied to all boundaries in the full-field and test cases. Both domains are subjected to
a harmonic propagating wave forcing, characterized by a wave amplitude, direction and period, as shown in Table 1. The
chosen wave period (or wavelength) ensures that multiple wavelengths fit exactly within the domain length, avoiding potential
numerical issues associated with periodic boundary conditions. Wind forcing is not considered, allowing us to isolate the effects
of wave-ice interactions.

All simulations in this study are conducted over a $24\,\mathrm{h}$ period to allow sufficient time for potential thermodynamic processes
to develop and to facilitate a direct comparison between dynamic simulations and coupled dynamic and thermodynamic sim-
ulations. It is important to note that the current simulations do not account for the phase transition between grease ice and ice
floes, which will be the focus of future works.

## 3 Results

In this section, we present simulation results to demonstrate the dynamic model and the coupling between the dynamics
and thermodynamics models. Three simulations are compared: dynamics only, thermodynamics only, and fully coupled. The
analysis focuses on two key variables: sea ice thickness and shear viscosity.

### 3.1 Full-field case

The propagating waves cause spatially heterogeneous variations in the shear viscosity field of grease ice and ice floes, as shown
in the snapshots after $24\,\mathrm{h}$ (Fig. 6). These variations reflect both the intrinsic differences in their rheological laws and the local
thickness changes induced by wave forcing, which also implicitly modulates viscosity. The influence of wave direction and

period on shear viscosity is also visually evident, and the time series of the domain-averaged values presented in Fig. 7 quantify these differences.

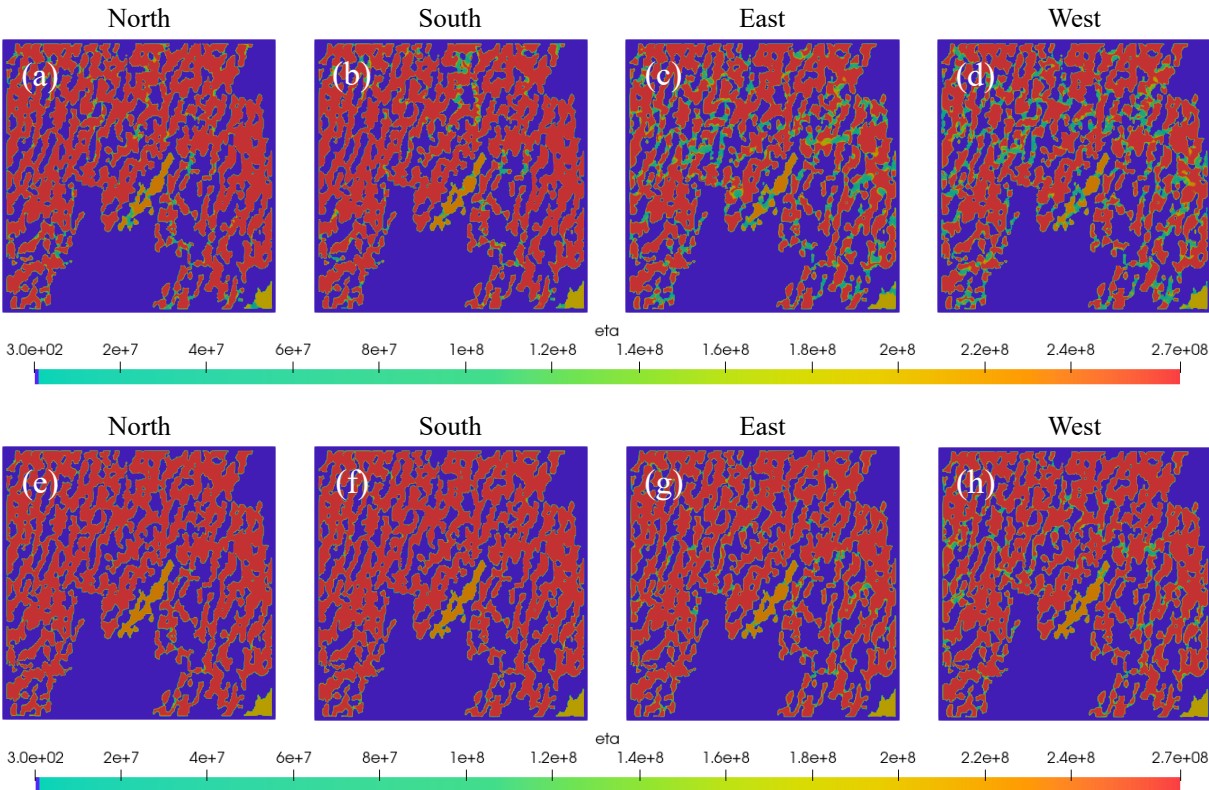

**Figure 6.** Sea-ice shear viscosity (in $\mathrm{kg\,s^{-1}}$) after $24\,\mathrm{h}$ from the full-field configuration for waves with different directions (from the north, east, south or west) and two selected wave periods: (a-d) $T = 8.8\,\mathrm{s}$, and (e-h) $T = 15.2\,\mathrm{s}$.

The shorter wave period, $T = 8.8\,\mathrm{s}$, results in a higher number of regions with lower sea ice shear viscosity ($< 1.4{\cdot}10^8\,\mathrm{kg\,s^{-1}}$) compared to the longer wave period. This is attributed to higher strain rates, consistent with the sea ice rheology (Eq. (12) and (13)). Lower viscosities are predominantly observed in regions where the ice floes are narrow, while larger ice floe regions remain largely unaffected by variations in wave period. The largest reduction in shear viscosity is observed with changing the wave direction; waves from the north and south are similar to each other but different from the east-west directions. This highlights the role of the orientation of narrow connections between ice floes relative to the wave direction in determining the sea ice viscosity, which is discussed in detail with the test cases in Section 3.2.

The time series of the domain-averaged sea ice shear viscosity, shown in Fig. 7, are dominated by the larger ice floe values. The curves show a small negative trend with high-frequency oscillations in the short-wave cases; which are also further analysed in Section 3.2. The simulations in the north-south directions show the highest values, with less sensitivity to wave period. In contrast, the different wave periods result in approximately a $20\,\%$ difference in the east-west case.

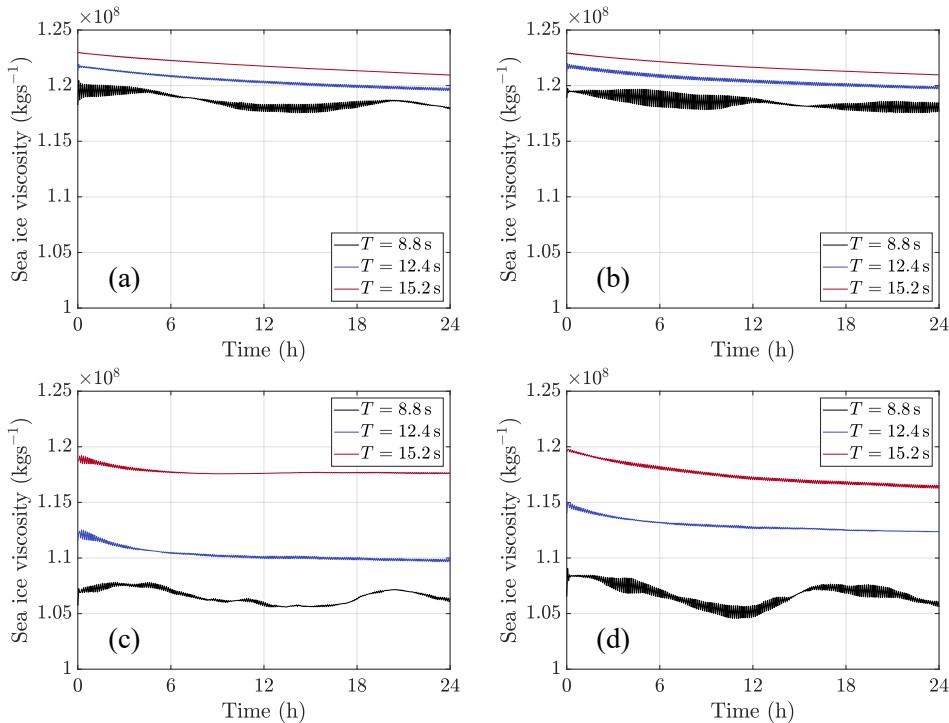

**Figure 7.** Domain-averaged sea ice shear viscosity (in $\mathrm{kg\,s^{-1}}$) for three different wave periods, with the wave direction coming from (a) the north, (b) the south, (c) the east, and (d) the west.

The role of heterogeneity, indicated by different percentages of grease ice and ice floes, is analysed in Fig. 8, where we partitioned the domain into 36 subdomains, each identified by a unique number, and calculated spatial and temporal averages of viscosity in relation to the ratio between ice floes and the grease ice. We focused on the lowest wave period because it shows the highest variability, and selected the south and west directions because of the similarity of the results for opposite directions. The time series for every subdomain are shown in Fig. A1 and A2 in Appendix A, in which we observe that the shorter wave periods exhibit a highly dynamic response that changes substantially between the subdomains and with respect to the domain-averaged results in Fig. 7.

The colour distributions in the heat maps of the ice floes percentage and viscosities (Fig. 8(b-d)) reveal the same pattern. The variation in sea ice viscosity is primarily determined by the percentage of ice floes within each subdomain, determining the overall magnitude of the mean viscosity.

Figure 8(e) and (f) illustrate the relative difference in shear viscosity between the highest and lowest wave periods. The viscosity values in the case with waves from the west are significantly higher than those from the south, indicating an increased sensitivity to wave periods. As previously mentioned, this discrepancy is attributed to the presence and orientation of the narrow connections and not just to the percentage. This behaviour is further supported by the subdomains with low relative differences,

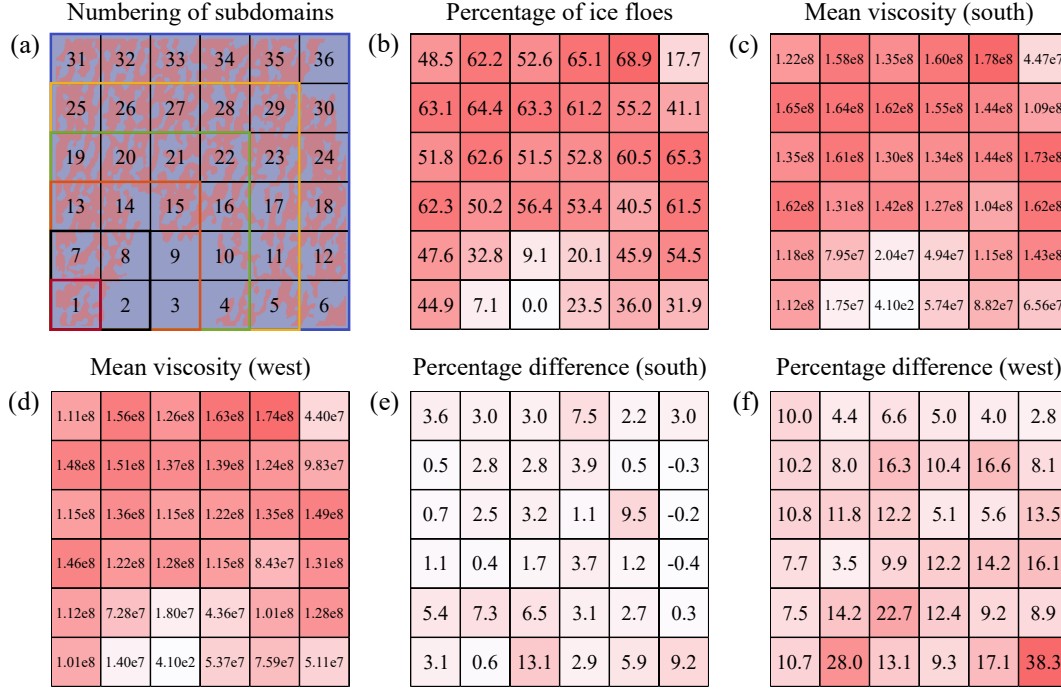

**Figure 8.** (a) Subdomain partitioning and reference numbers, with colours indicating one example combination for each grid size ; (b) prescribed percentage of ice floes in each subdomain; (c) the mean sea ice shear viscosity (in $\mathrm{kg\,s^{-1}}$) in each subdomain over a $24\,\mathrm{h}$ period for waves propagating from the south with a wave period of $T = 8.8\,\mathrm{s}$, (d) same as (c) but for waves propagating from the west; (e) percentage difference in viscosity between wave periods of $T = 15.2\,\mathrm{s}$ and $T = 8.8\,\mathrm{s}$ for waves propagating from the south, and (f) same as (e) but for waves propagating from the west.

which appear in the rightmost panels of Fig. 8(e) and the top panels of Fig. 8(f). These subdomains are dominated by larger ice floes with fewer narrow connections, making them less responsive to the different wave periods.

The results of these simulations can be further summarized in Fig. 9, where we observe an emergent linear response of the mean viscosity of each subdomain to the percentage of ice floes. The domain is limited to $69\%$, as this is the maximum value observed in the SAR image (Fig. 8(b)), and the intercept at $0\%$ represents the viscosity of grease ice ($\approx 440\,\mathrm{kg\,s^{-1}}$). The north-south orientation of the incoming wave describes a linear relationship between sea ice viscosity and the percentage of ice floes in each subdomain (see the equation in Fig. 9). This relationship slightly deviates when all data points are included, due

to the increased spread at higher floe percentages, which is clearly dependent on both wave direction and period. This indicates that there is a further relationship between the pattern of the simulated heterogeneous field and the wave direction, which adds to the influence of the floe percentage.

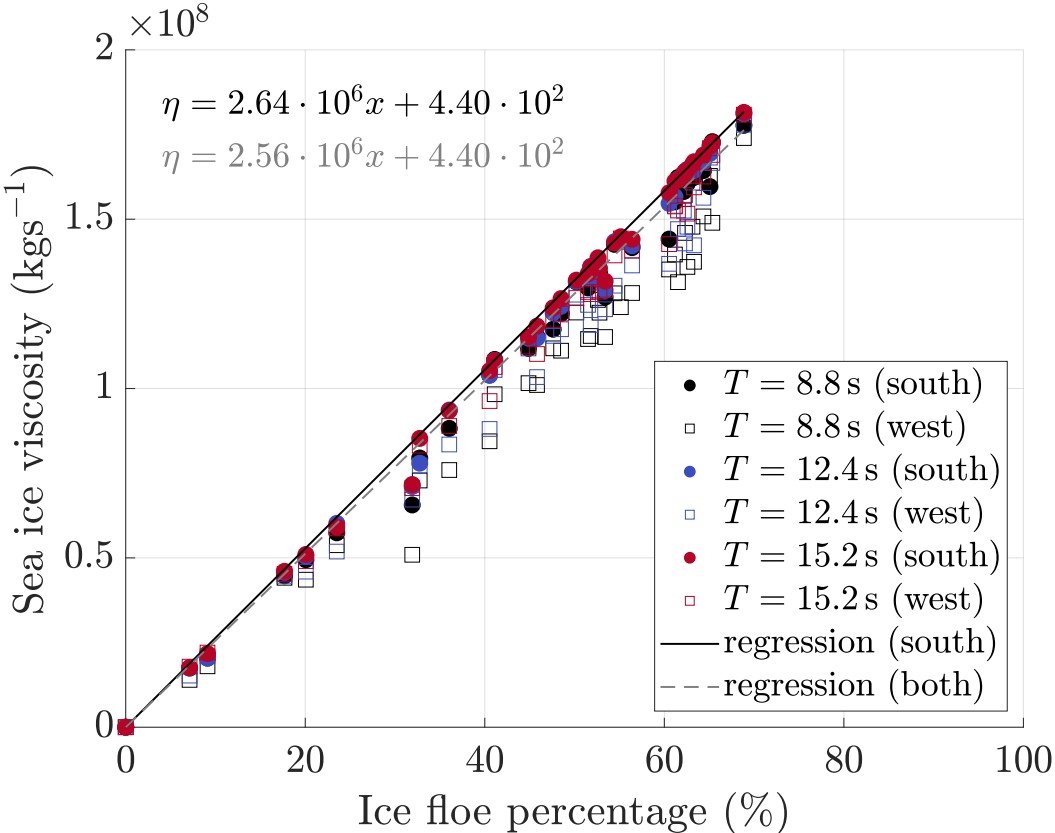

**Figure 9.** Linear regressions of the mean shear viscosity (in $\mathrm{kg\,s^{-1}}$) for the subdomains in Fig. 8(a) against the percentage of floes in each subdomain. The symbols represent different wave conditions. The intercept at $0\,\%$ ice floes corresponds to grease-ice viscosity ($\approx 440\,\mathrm{kg\,s^{-1}}$), as indicated by the regression equations shown in the top-left corner.

### 3.1.1  Scales of heterogeneity

We have performed a scaling analysis on the full-field results to evaluate the presence of a scaling law, which is often found in sea ice kinematics (e.g. Weiss, 2013). In this case, we are interested in the response of viscosity to the spatial scales of heterogeneity, here represented by the combination of interstitial grease ice and ice floes. Therefore, we considered grid sizes of increasing size from $1 \times 1$ to $6 \times 6$, with the latter corresponding to the full domain (see Fig. 8(a)) and calculated the mean shear viscosity and standard deviation in all possible group combinations.

The results are presented in Table 3, showing a strong scale invariance of the mean viscosity from $840 \times 840\,\mathrm{m^2}$ for a $1 \times 1$ grid up to $5040 \times 5040\,\mathrm{m^2}$ for a $6 \times 6$ grid. The 840 m scale is already sufficient to capture the heterogeneity of the ice cover, and variations in ice type patterns do not affect the mechanical response at the larger scales up to approximately 5 km. While variance increases with length scale, the average viscosity is well reconstructed, as the 10 m resolution provides a good estimate of the percentage of heterogeneity.

This behaviour is independent of the wave direction, as also shown in Table 3. The direction changes the absolute value, but we can conclude that the domain-averaged sea ice viscosity is primarily controlled by the ratio of ice floes to grease ice, which is also scale-invariant in this configuration.

Based on the inclusion of smaller scale processes that we assume realistic, the emergence of the linear relationship presented in Fig. 9 and the strong scale invariance of the mean viscosity of sea ice, we are confident that our results can be used to extract
properties at larger scales as further discussed in Section 4.

**Table 3.** Scaling analysis, where **size**: tile group size, **combinations**: possible combinations, **N**: total number of subdomains, **viscosity S**: mean sea ice viscosity (south), **std S**: standard deviation (south) **viscosity W**: mean sea ice viscosity (west), **std W**: standard deviation (west).

| size | combinations | N | viscosity S ($\mathrm{kg\,s^{-1}}$) | std S ($\mathrm{kg\,s^{-1}}$) | viscosity W ($\mathrm{kg\,s^{-1}}$) | std W ($\mathrm{kg\,s^{-1}}$) |
|------|-------------|---|------------|-------|------------|-------|
| $1 \times 1$ | $1 \times 36$ | 36 | $1.19 \times 10^8$ | $4.72 \times 10^7$ | $1.07 \times 10^8$ | $4.37 \times 10^7$ |
| $2 \times 2$ | $4 \times 25$ | 100 | $1.21 \times 10^8$ | $3.65 \times 10^7$ | $1.09 \times 10^8$ | $3.31 \times 10^7$ |
| $3 \times 3$ | $9 \times 16$ | 144 | $1.22 \times 10^8$ | $2.65 \times 10^7$ | $1.10 \times 10^8$ | $2.42 \times 10^7$ |
| $4 \times 4$ | $16 \times 9$ | 144 | $1.22 \times 10^8$ | $1.87 \times 10^7$ | $1.10 \times 10^8$ | $1.78 \times 10^7$ |
| $5 \times 5$ | $25 \times 4$ | 100 | $1.21 \times 10^8$ | $9.34 \times 10^6$ | $1.09 \times 10^8$ | $9.73 \times 10^6$ |
| $6 \times 6$ | $36 \times 1$ | 36 | $1.19 \times 10^8$ | 0 | $1.07 \times 10^8$ | 0 |

### 3.1.2   Dynamics and thermodynamics coupling

The coupling between the dynamics and thermodynamics models is examined using the full-field case, with the initial layout illustrated in Fig. 5(a). We classify both ice floes and grease ice into six distinct thickness categories:

Ice floes:   $0 - 0.10\,\mathrm{m}$,   $0.10 - 0.12\,\mathrm{m}$,   $0.12 - 0.14\,\mathrm{m}$,   $0.14 - 0.16\,\mathrm{m}$,   $0.16 - 0.18\,\mathrm{m}$,   $0.18 - 0.28\,\mathrm{m}$.

Grease ice:   $0 - 0.06\,\mathrm{m}$,   $0.06 - 0.09\,\mathrm{m}$,   $0.09 - 0.12\,\mathrm{m}$,   $0.12 - 0.15\,\mathrm{m}$,   $0.15 - 0.18\,\mathrm{m}$,   $0.18 - 0.24\,\mathrm{m}$.

The upper limits of $0.28\,\mathrm{m}$ for ice floes and $0.24\,\mathrm{m}$ for grease ice are considered sufficient, as significant thickening beyond these thresholds is unlikely within a $24\,\mathrm{h}$ period.

Figure 10 presents the spatial distribution of sea ice thickness and shear viscosity at $t = 24\,\mathrm{h}$ for waves propagating from
the west with a period of $T = 8.8\,\mathrm{s}$, comparing results from the dynamics-only model, the thermodynamics-only model, and the fully coupled dynamics & thermodynamics model (see also the videos in the supplementary material in the Appendix).

As previously mentioned in Section 2.3, the spatial distribution of the thermodynamics-only model is obtained by using the coupled model with wave amplitude set to zero, thereby excluding sea ice dynamics while preserving the correct category proportions. As a result, Fig. 10(b) and (f) exhibit a uniformly increased ice thickness and shear viscosity compared to the
dynamics-only results in Fig. 10(a) and (e). This uniformity arises from the absence of ice motion, which prevents interaction and, consequently, the redistribution of sea ice thickness and viscosity between the two ice types. The increase in thickness, as illustrated by the difference between the fully coupled model and the dynamics-only model in Fig. 10(d), is most pronounced

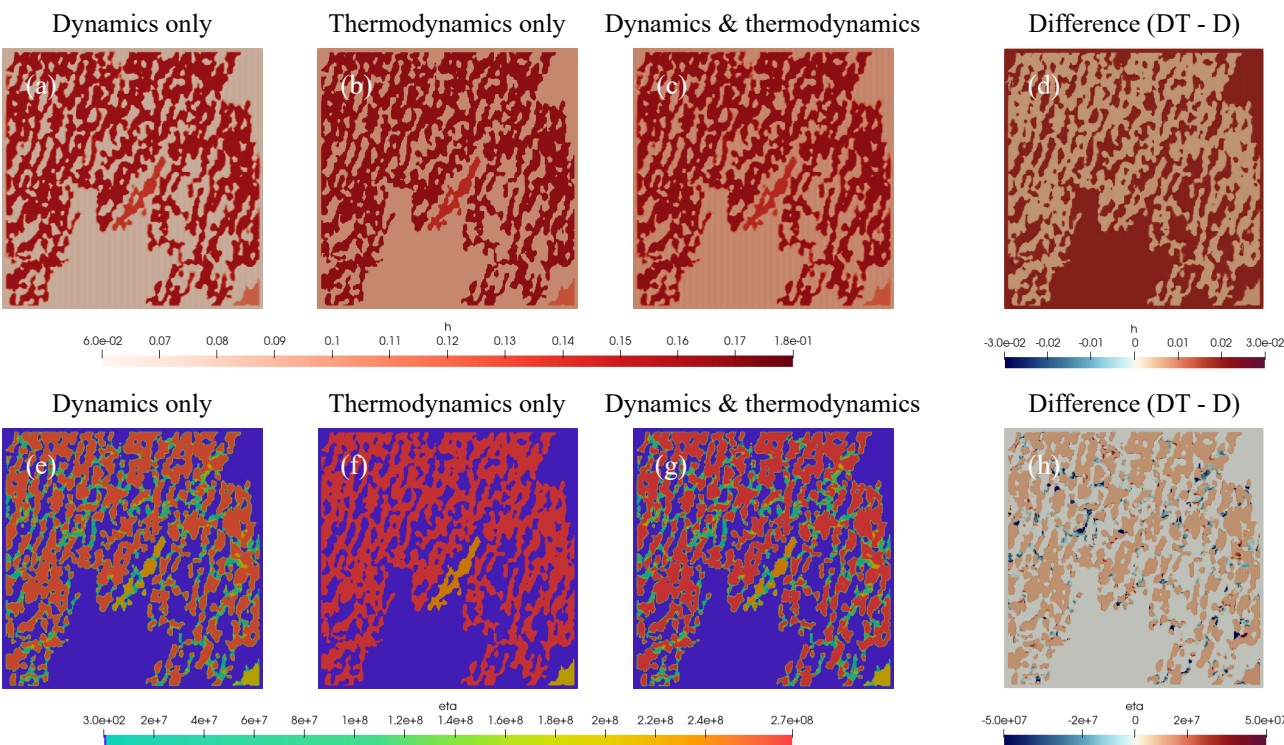

**Figure 10.** (a-d) sea ice thickness (in m), and (e-h) sea ice shear viscosity (in $\mathrm{kg\,s^{-1}}$) for waves originating from the west with a wave period of $T = 8.8\,\mathrm{s}$, comparing the outcomes of (a, e) the dynamics-only model, (b, f) the thermodynamics-only model, and (c, g) the fully coupled dynamics & thermodynamics model. Panels (d) and (h) show the difference in sea ice thickness and shear viscosity between the fully coupled model and the dynamics-only model.

in the grease ice regions due to the higher growth rate of grease ice compared to ice floes. However, this increase is not visible in the viscosity distributions due to the chosen colour bar, which emphasizes the ice floes that dominate the overall viscosity
within the domain.

When comparing the shear viscosity fields from the dynamics-only model in Fig. 10(e), and the fully coupled dynamics & thermodynamics model in Fig. 10(g), we observe that the spatial distribution of lower-viscosity regions, highlighted in green, remains nearly identical. However, the difference plot in Fig. 10(h) shows that, in some of these regions, the dynamics-only model exhibits higher viscosities (indicated by negative values), despite the increased thickness in the fully coupled model.
This highlights the importance of dynamics in shaping the viscosity distribution of sea ice, as the interaction between dynamics and thermodynamics is inherently nonlinear.

Figure 11(a) and (b) illustrate the comparison of the domain-averaged sea ice thickness and shear viscosity time series over a $24\,\mathrm{h}$ period between the dynamics-only model and the fully coupled dynamics & thermodynamics model for waves propagating
from the west with a period of $T = 8.8\,\mathrm{s}$.

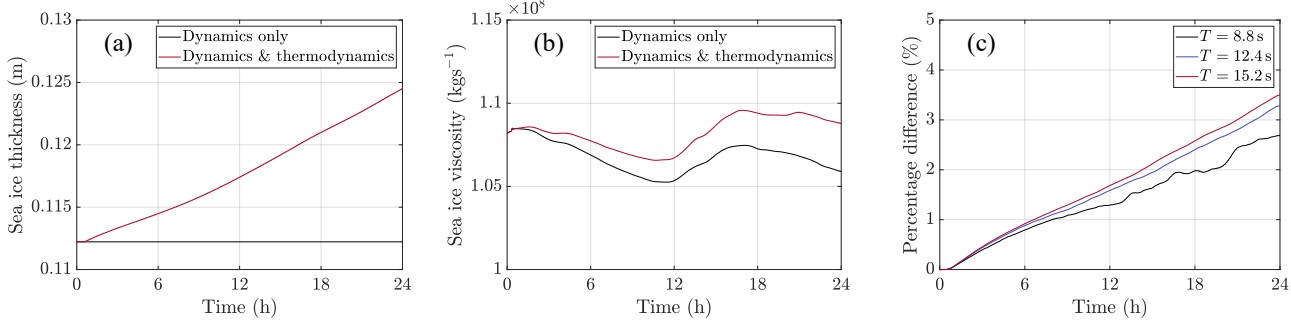

**Figure 11.** Comparison between the dynamics-only model and the fully coupled dynamics & thermodynamics model, showing the domain-averaged of (a) sea ice thickness (in m), and (b) sea ice shear viscosity (in $\mathrm{kg\,s^{-1}}$) for waves originating from the west with a wave period of $T = 8.8\,\mathrm{s}$. Panel (c) displays the percentage difference between the curves in panel (b), as well as for wave periods, $T = 12.4\,\mathrm{s}$ and $T = 15.2\,\mathrm{s}$, which are not shown in panel (b).

Initially, both models produce identical results for the domain-averaged sea ice thickness and shear viscosity, reflecting the spin-up time required for the dynamic model to reach equilibrium conditions. The spin-up time, equivalent to five times the wave period, marks the phase before the thermodynamic model is incorporated, as illustrated in Fig. 3 in Section 2.3.

In the dynamics-only model, sea ice thickness remains constant over time, as mass conservation is inherently preserved. In contrast, the dynamics & thermodynamics model shows an increase in sea ice thickness due to thermodynamic ice growth. Over a $24\,\mathrm{h}$ period, the domain-averaged sea ice thickness, accounting for both ice floes and grease ice, increases by over $1\,\mathrm{cm}$. The domain-averaged sea ice thickness results are nearly identical for the two longer wave periods. This similarity arises from the identical heat fluxes applied across all scenarios.

A similar trend is observed in the domain-averaged shear viscosity, as depicted in Fig. 11(b), with viscosity increasing due to thermodynamic sea ice growth. Despite the difference in viscosity values, both viscosity curves exhibit a comparable shape, likely attributed to the similar spatial distribution of lower-viscosity regions in both cases, as discussed previously. Figure 11(c) illustrates the relative difference, expressed as a percentage, between the domain-averaged shear viscosity for the dynamics-only model and the dynamics & thermodynamics model across three wave periods, $T = 8.8\,\mathrm{s}$, $T = 12.4\,\mathrm{s}$ and $T = 15.2\,\mathrm{s}$. The relative difference increases slightly with the wave period.

The relationship between the dynamics and thermodynamics models is subsequently examined using the method described in Section 2.3 and Eq. (26), with the same configuration as above. Figure 12(a) and (b) present a comparison of the domain-averaged sea ice thickness and shear viscosity, contrasting the decoupled dynamics and thermodynamics models with the fully coupled dynamics & thermodynamics model.

A negligible discrepancy is observed in the domain-averaged sea ice thickness, as depicted in Fig. 12(a). In contrast, the difference in domain-averaged sea ice shear viscosity between the decoupled and coupled models (Fig. 12(b)) grows over time, with a relative difference of $1.3\%$ at $t = 24\,\mathrm{h}$ (Fig. 12(c)). This suggests that the decoupled configuration overestimates the

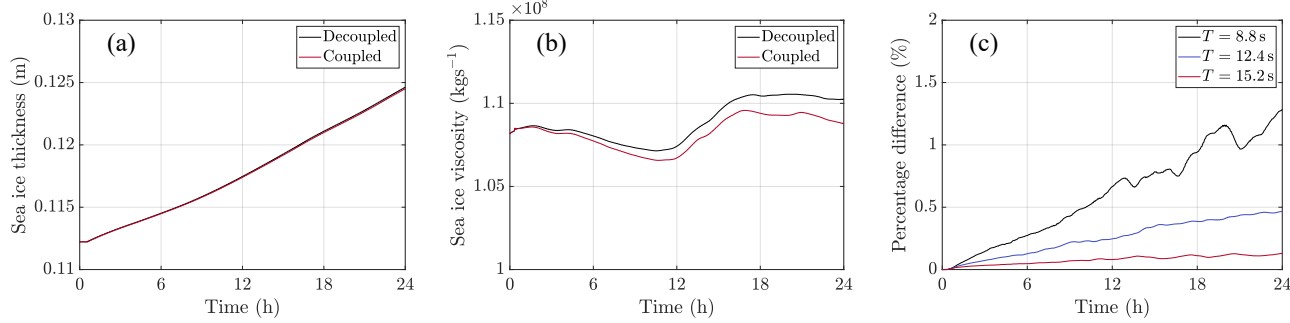

**Figure 12.** Comparison between the decoupled dynamics and thermodynamics models and the fully coupled dynamics & thermodynamics model, showing the domain-averaged of (a) sea ice thickness (in m), and (b) sea ice shear viscosity (in $\mathrm{kg\,s^{-1}}$) for waves originating from the west with a wave period of $T = 8.8\,\mathrm{s}$. Panel (c) displays the percentage difference between the curves in panel (b), as well as for wave periods, $T = 12.4\,\mathrm{s}$ and $T = 15.2\,\mathrm{s}$, which are not shown in panel (b).

domain-averaged sea ice viscosity. Notably, the relative difference diminishes with increasing wave period, decreasing from $1.3\,\%$ to $0.1\,\%$ at $t = 24\,\mathrm{h}$. Over longer timescales, on the order of five to ten days, these differences will accumulate, potentially resulting in a more substantial difference between the decoupled and coupled models.

## 3.2 Test cases

In this section, we present a detailed analysis of the idealised configuration introduced in Section 2.4 describing two larger floes of different thicknesses connected by a narrow bridge. This configuration is designed to help interpret and better understand the results obtained in the full-field case. This case also shows the response of the grease ice component to the wave that was not visually evident in Section 3.1 due to the larger scale. The focus will be on the role of floes' orientation and connectivity in relation to the wave direction.

The model is able to describe the features of grease ice thickness and its interaction with the ice floes under different wave conditions, as indicated by the darker shading - which changes position depending on the orientation and wave period - near the interface between the grease ice and ice floes (see Fig. 13(a-h) and the supplementary videos in the Appendix).

Figure 13(i-p) focus exclusively on the shear viscosity of the two ice floes and the narrow connection between them. The viscosity of the two ice floes is unaffected by the wave period, as both sets of figures display similar colour patterns within the ice floes. In contrast, the viscosity of the narrow connection depends on both the wave period and its orientation relative to the wave. When the connection is aligned perpendicular to the wave front, the viscosity is minimally affected, whereas it reaches its minimum when the connection is parallel to the wave front.

If we now calculate the spatial averages as done for the full-field case in Section 3.1, we observe a similar response on the mean shear viscosity. Figure 14 shows substantial differences between the horizontal and vertical orientations in panels (a) and (b), which correspond to the bridges aligned perpendicular and parallel to the wave front. We notice that the absolute values and the relative changes in viscosity are scaled to the size of this test-case configuration, and to the shape and proportion of ice

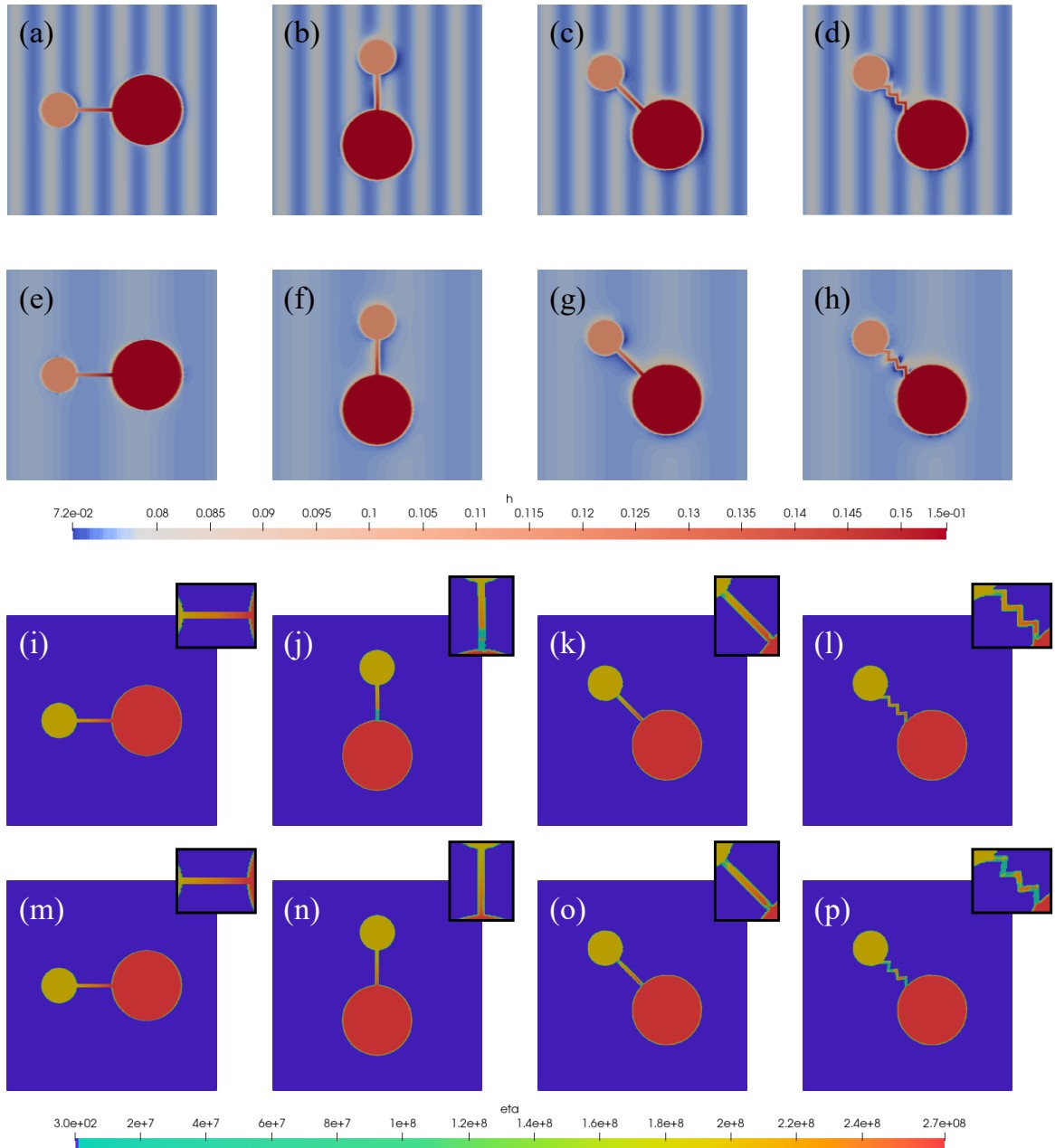

**Figure 13.** (a-d) sea ice thickness (in m; the colour bar ranges are chosen to emphasize the wave motion in grease ice), and (i-l) sea ice shear viscosity (in $\mathrm{kg\,s^{-1}}$) for waves with a wave period of $T = 8.8\,\mathrm{s}$. (e-h) sea ice thickness, and (m-p) sea ice shear viscosity for waves with a wave period of $T = 15.2\,\mathrm{s}$. The black box in the top-right corner highlights the narrow connection between two ice floes.

floes in the domain. This configuration has a larger proportion of grease ice (see Fig. 5), which was chosen as a compromise to
illustrate both the response of the grease ice and the role of ice floes' orientation. In the orientation perpendicular to the wave

front (Fig. 14(a)), the domain-averaged sea ice viscosity is unaffected by the different wave periods because the difference between the curves is negligibly small. This can be attributed to the orientation, as both floes (including the bridge) behave more as a rigid body, making the system less sensitive to the wave period. In this configuration, the bridge primarily experiences compression and tension. In contrast, when the bridge is parallel to the wave direction (Fig. 14(b)), the floes can move more independently, creating a higher strain rate in the bridge and, consequently, a lower viscosity. As a result, we observe a greater separation between the curves with the largest wave period corresponding to the highest viscosity. This explains why different responses to the wave periods are observed across subdomains, as illustrated in Fig. A1 and A2. These test-case results also demonstrate that the shape and orientation of the bridge influence the oscillations in the viscosity curves shown in Fig. 7 and in Fig. A1 and A2.

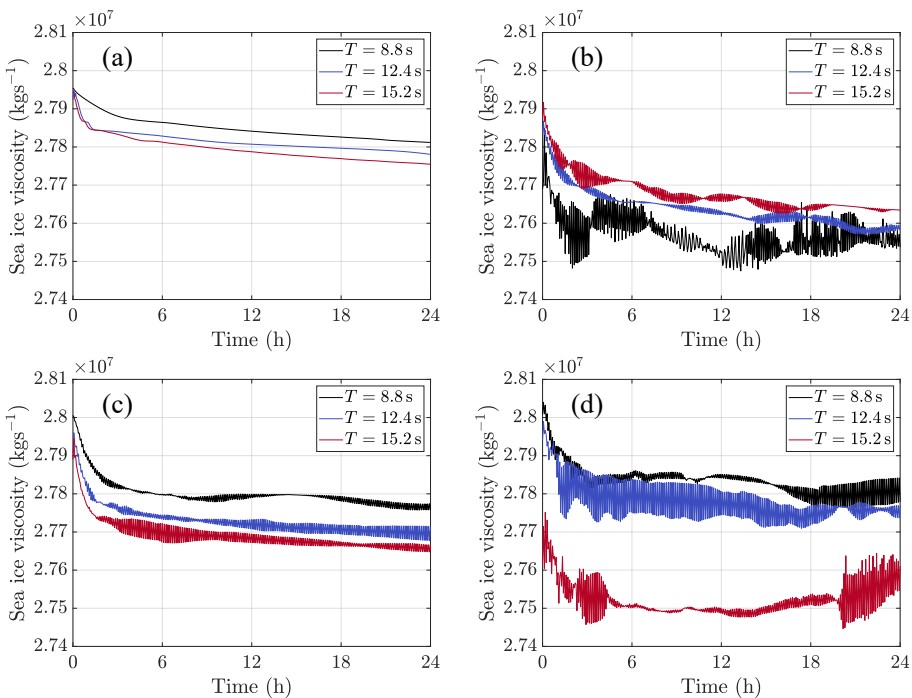

**Figure 14.** Domain-averaged sea ice shear viscosity (in $\mathrm{kg\,s^{-1}}$) for three different wave periods, with the narrow connection in (a) horizontal, (b) vertical, (c) diagonal, and (e) zigzag direction.

Figure 14(c) and (d) present the results for the inclined configuration. The straight inclined bridge is more similar to the results shown in Fig. 14(a), while Fig. 14(d) exhibits more pronounced oscillations that are also observed in some subdomains of the full-field configuration (Fig. A1 and A2). These oscillations, which are of negligible magnitude for this test-case configuration, are more pronounced in the full-field subdomains. We attribute this to the shape of the zigzag connection, which adds complexity to the shear dynamics. While noise from numerical interface approximations cannot be excluded, the possible

emergence of internal resonance - due to the domain periodicity and wavelength selection - as well as harmonics amplified in the full-field case also requires further investigation.

## 4   Discussion

The proposed model, WIce-FOAM 1.0, is designed to explore the dynamic response of the heterogeneous sea-ice cover, composed of consolidated ice floes and interstitial grease ice, to harmonic wave forcing. The rationale for this study originates

from the need to describe the characteristics of the Antarctic marginal ice zone, which is a mosaic of pancake ice floes cemented together (e.g. Wadhams et al., 2018; Alberello et al., 2019; Vichi, 2022). Our results highlighted that the heterogeneity of the sea-ice surface, and not only the thickness, alters the overall mechanical response of sea ice to wave stresses. The spatially averaged viscosity field is sensitive to the wave direction and the period, and the intensity of the response depends on the percentage of solid floes. We found a lower shear viscosity in the regions where the floes aggregate in narrower formations.

Through selected test cases, we found that the presence and orientation of the narrow junctions between ice floes relative to the wave direction play a key role in the mean viscosity of an area of heterogeneous sea ice. We also observed changes in the magnitude and oscillatory behaviour of the mean shear viscosity that are indicative of resonance at smaller scales that can be propagated to the kilometre scales.

     Our approach is agnostic, and we deliberately did not consider in our simulations the likely existence of an underlying

relationship between the patterns of heterogeneity and the direction of the waves. It is important to remember that these patterns are not sea-ice bands (e.g. Saiki et al., 2021), since this is not an open drift condition and the scales are smaller than the 10 km usually observed for the bands. Nevertheless, we observe a clear anisotropy in the response of the shear viscosity to the wave direction. We removed the wave patterns in Fig. 4, which indicated a propagation from west-northwest, and we do observe a series of bands oriented perpendicularly to this direction (see Fig. 5(a)). The bands may have been arranged this way

by the wave motion, and we notice that the narrow bands show a lower viscosity with waves coming from the west (Fig. 6(d) and (h)).

     The addition of thermodynamic processes alters the viscosity of sea ice in the dynamic model by $3\%$ at $t = 24\,\mathrm{h}$. This small percentage increases slightly with an increase in the wave period. However, the coupling between dynamics and thermodynamics exhibits a nonlinear response that grows over time. Despite the detailed formulation of the fully coupled model, the

dynamics requires a small time step of less than a second, resulting in significant computational costs. While differences observed over a $24\,\mathrm{h}$ period remain limited, they accumulate over time. This indicates that as computational capabilities improve, enabling extended simulations over a longer period, the impact of these differences will become increasingly apparent. However, given that environmental and waves' characteristics may change with the storm scales of a few days, this nonlinearity may not grow indefinitely. In addition, the lack of phase change between grease ice and solid floes limits the length of our sim-

ulations, since grease ice grows faster than the solid floes and would become thicker. Some degree of compaction or interaction with the rims of the ice floes is expected to occur, but there are no direct observations of this process.

One major outcome of our approach is an emerging linear relationship between the mean shear viscosity and the ice floe percentage within each subdomain (Fig. 9). Several test cases considering different shapes and sizes of the solid floes were implemented during the onset of this work, and led to responses of the mean viscosity field that were of difficult interpretation. This is demonstrated by the figures in the Appendix, in which different configurations lead to different responses of the mean viscosity to the waves characteristics. Only the use of the pseudo-realistic full-field configuration from an SAR image allowed us to discriminate emerging patterns. There is not enough knowledge to accurately determine the type of ice from radar intensity, and we used a threshold derived from the field itself to arbitrarily assign rheology and thickness to the ice field. We do not imply generalization to all SAR images, but we suggest that this model can already be used to determine the scales of heterogeneity and inform the design of parameterisations that include the effects of waves on sea-ice mechanics. Without explicitly resolving for the waves, it should be possible to derive a parameterisation of the heterogeneous rheology based on the percentage of young ice. Current sea ice models do not yet implement a deterministic calculation of young ice, but a few parameterisations are already included to resolve polynya conditions (Fang et al., 2024). Our scaling analysis indicates that the percentage of ice floes can be obtained from SAR images at the scales of current sea ice models, and used to derive a mean viscosity value that accounts for the wave action based on this emergent linear relationship. The wave direction and the period partly affect the magnitude, and we notice in Fig. 9 that there is a likely range between $30 - 70\,\%$ over which this happens. Our range of values is limited to $69\,\%$, but we expect that at $100\,\%$ floe coverage, the viscosity would converge to the maximum extrapolated value of $2.6 \cdot 10^8\,\mathrm{kg\,s^{-1}}$ from Eq. (13).

We acknowledge that this model contains major assumptions and compromises due to the limitations of our current knowledge on these processes in the Antarctic and the chosen computational framework. The model does not include leads because it focuses on regions where the cover is $100\,\%$ but still thin, and thus is affected by the penetration of waves. A multiphase approach involving more than two phases cannot be implemented in OpenFOAM without abandoning the Volume of Fluid (VOF) technique currently used, the IsoAdvector method. This would degrade the representation of the interface between interstitial grease ice and consolidated ice floes. For this preliminary study, which builds on the work of Marquart et al. (2021, 2023), we preferred to preserve the interfaces of the mosaic, but we do not exclude a further development to include a seawater phase and the phase transition between water and the different types of sea ice. Our results have highlighted the importance of the floes' percentage in determining the mechanical response to the waves, and therefore other more diffusive methods that ensure mass conservation at the expense of details can be considered in the future.

Another relevant assumption is that the rheology of both constituents is known, with viscous grease ice and viscous-plastic solid floes (see Section 2.1.2). This latter method is the standard parameterisation proposed by Hibler III (1979) used in most sea-ice models, which do not account for the complex stratigraphy and the prevalence of granular structure observed in Antarctic floes (e.g. Lytle and Ackley, 2001; Skatulla et al., 2022; Tison et al., 2020; Johnson et al., 2023). There are no recent works on the compressive or shear response of granular ice (Paul et al., 2023), and the literature, as well as the numerical models, assume that sea ice is columnar in its structure. Finally, we only considered waves to be harmonic and unidirectional, and ignored the attenuation. This is justified by the duration of our simulations in terms of time; for this reason, we considered experiments with thin ice, which also allowed thermodynamic effects to become apparent within a one-day time scale.

## 5    Conclusions

We presented a modelling framework, WIce-FOAM 1.0, developed in OpenFOAM to explicitly resolve the dynamic and thermodynamic processes in sea ice composed of two distinct ice types at the sub-kilometre scale. In our model, cells identified as ice floes or grease ice may contain both ice types, though one type dominates. This model serves as an initial testing platform to explore the response of heterogeneous thin ice to the effects of waves. It is a useful tool for designing experimental in situ and laboratory setups and for deriving emerging relationships to inform the parameterisation of larger-scale numerical sea ice models. We demonstrated its functionality through both realistic and idealised simulations based on Antarctic sea ice examples, where the floes consist of agglomerated pancake ice and the interstitial component is grease ice. However, the model can be applied to any sea ice configuration where solid floes and interstitial ice coexist. Despite being constrained by limited computational time scales, our initial results show an emerging linear relationship between the fraction of solid floes across multiple spatial scales and the mean shear viscosity. The mean viscosity is affected by the direction and period of the incident waves, which, in principle, would require sea ice models to explicitly resolve wave features. However, our findings suggest the possibility to parameterise the mechanical response of heterogeneous sea ice to waves, even without their explicit representation, assuming the model can incorporate wave characteristics and simulate the presence of interstitial ice. Since our work assumes unattenuated waves, further research is required to include the feedback between the penetrating waves and sea ice, as well as the phase transition between interstitial grease ice and solid floes.

**Appendix A**

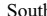

**Figure A1.** Spatially-averaged sea ice shear viscosity (in $\mathrm{kg\,s^{-1}}$) over a 24 h period across 36 subdomains, for three different wave periods, with wave propagation from the south. The black, blue and red curves correspond to wave periods of $T = 8.8\,\mathrm{s}$, $T = 12.4\,\mathrm{s}$, and $T = 15.2\,\mathrm{s}$, respectively. In most panels, shear viscosity is the highest for the longer wave periods and the lowest for the shorter. The shortest wave period exhibits a more dynamic response, which changes substantially between the subdomains and with respect to the domain-averaged results in Fig. 7. Panel number 3 consists solely of grease ice, resulting in shear viscosity values five orders of magnitude smaller than those of the adjacent panels. A clear trend is observed between grease ice viscosity and wave period, with the shear viscosity increasing as the wave period increases. Moreover, we observe that the spatially-averaged shear viscosity of grease ice is independent of the wave direction, as the results from panel 3 are the same as in Fig. A2.

West

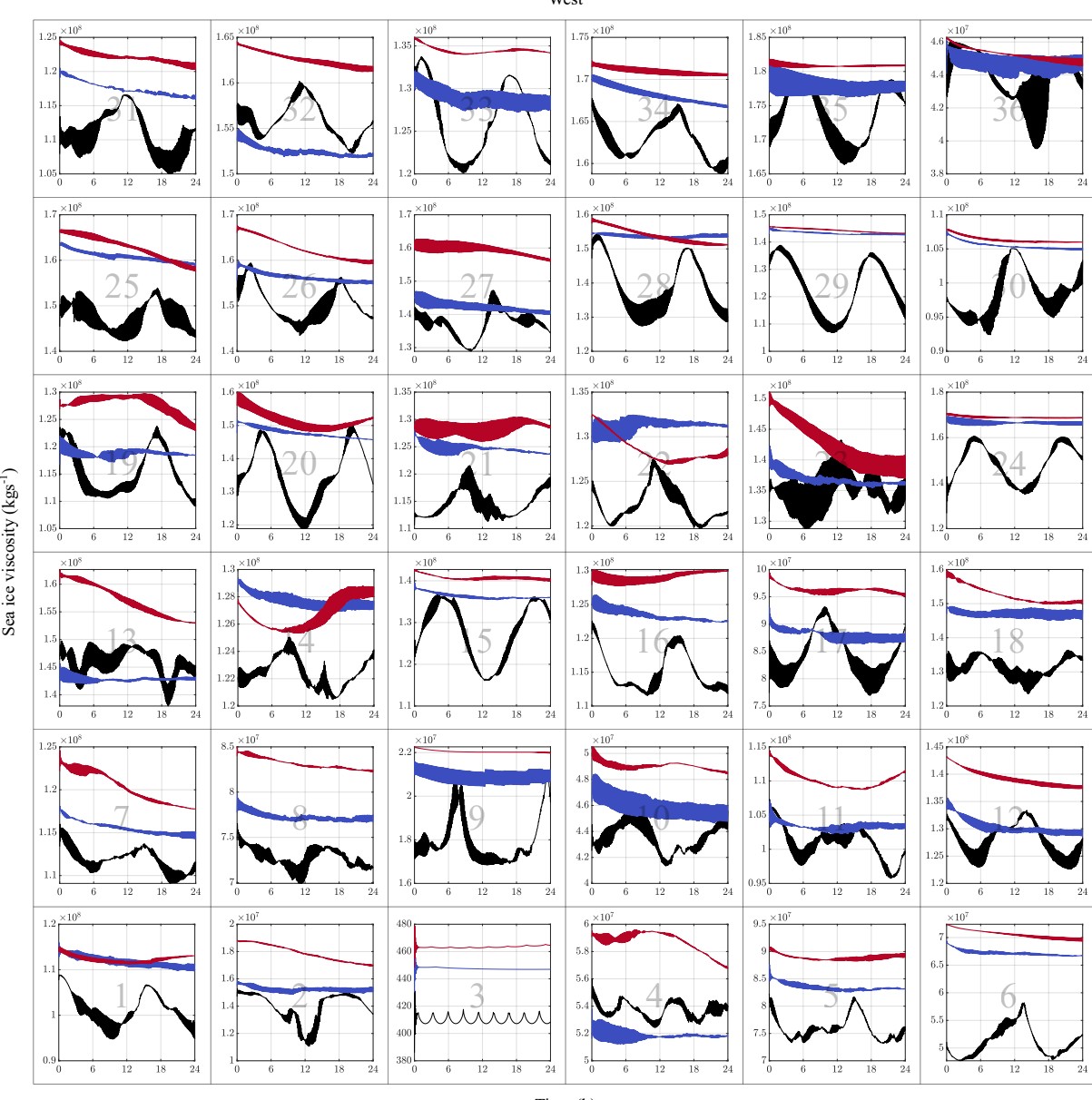

Sea ice viscosity (kgs⁻¹)

Time (h)

**Figure A2.** As in Fig. A1 but for wave propagation from the west.

*Code and data availability.* The simulation and solver files of WIce-FOAM 1.0 - which include both the fully coupled dynamic model implemented in OpenFOAM-v2306 and the thermodynamic model in Python - are freely available on Zenodo: https://doi.org/10.5281/ zenodo.16681435. The ERA5 datasets (Hersbach et al., 2018a, b) used in the thermodynamic model are freely available online at the Copernicus Climate Data Store (https://cds.climate.copernicus.eu/datasets/reanalysis-era5-single-levels?tab=download), or can be accessed via the Zenodo link provided above.

*Video supplement.* Test case: four videos show the evolution of thickness and viscosity fields for two larger floes connected by a narrow horizontal and vertical bridge, aligned perpendicular and parallel to the wave front. The animations correspond to Fig. 13(a), (b), (i) and (j) in Section 3.2. They depict the thickness and viscosity evolutions over a $24\,\mathrm{h}$ period and are played at an accelerated speed. Full-field case: two videos show the evolution of thickness and viscosity fields of the dynamics-only model, the thermodynamics-only model, and the fully coupled dynamics & thermodynamics model. The animations correspond to Fig. 10 in Section 3.1.2. They depict the thickness and viscosity evolutions over a $24\,\mathrm{h}$ period and are played at an accelerated speed. All videos are freely available on ZivaHub: https://doi.org/10.25375/uct.28956746.v1

.

*Author contributions.* Conceptualization: R.M. and M.V.; methodology: R.M. and M.V.; software: R.M. and A.B.; validation: all authors; writing-original draft preparation: R.M. and M.V.; writing-review and editing: all authors. All authors have read and agreed to the published version of the manuscript.

*Competing interests.* The authors declare that they have no conflict of interest.

*Acknowledgements.* The authors would like to thank Professor Arnaud Malan for his help and the fruitful discussions that contributed to the success of this research. Computations were performed using facilities provided by the University of Cape Town's ICTS High Performance Computing team: hpc.uct.ac.za.

*Financial support.* This work has received funding from the European Union's Horizon 2020 Research and Innovation programme under grant agreement no. 101003826 via the project CRiceS (Climate Relevant interactions and feedbacks: the key role of sea ice and Snow in the polar and global climate system).

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
