# Peer review of "WIce-FOAM 1.0: Coupled dynamic and thermodynamic modelling of heterogeneous sea ice and waves using OpenFOAM-v2306"

_EGUsphere, 2025_

## Author Comment (AC2)

**Comments from Referee #1, followed by the authors' responses**

**Referee #1** - This paper presents a novel sea ice model that represents the mixed phase between grease ice and consolidated ice floes, including both dynamic and thermodynamic components. The work investigates the response of this model to idealized wave forcing for a sea ice configuration derived from SAR imagery. Simpler configurations are also considered as sensitivity experiments. The key diagnostics are the mean viscosity and local strain rate as a function of wave properties and ice floe concentration relative to grease ice. This work represents a useful advance in modelling sea ice within marginal ice zones and its response to wave forcing. My specific comments are listed below.

**Referee #1** - Eq. 6: Does the vertical component of wave velocity play a role in these simulations? The rheology seems to be described for a 2-D simulation only (section 2.1.2).

Authors' response - The vertical component of wave velocity does not play a role in the simulations, however, for completeness, we report it in Eq. 6. We will clarify this in the revised manuscript.

**Referee #1** - L138: What does 'apical plane' mean?

Authors' response - The apical plane refers to the upper surface of the ice, which is exposed to the atmosphere and lies parallel to the $xy$-plane. We also recall that the basal plane refers to the lower surface of the ice, which is in contact with the ocean. We will clarify this in the revised manuscript.

**Referee #1** - Table 1: Can you describe how these values were chosen?

Authors' response - The parameter values in Table 1 can be organised into three categories. Ice floe rheology parameters are based on the model by Hibler III (1979) and related formulations, such as Mehlmann and Richter (2017). Notably, the limit of the effective strain rate (Leppäranta and Hibler III, 1985) was deliberately reduced by one order of magnitude to ensure numerical stability. While this adjustment influences the solidity of the ice floes by altering their viscosity, it was found that the overall simulation results remain unaffected. The values for grease ice rheology were derived from literature sources, such as Paul et al. (2021), and further refined through empirical tuning via iterative simulations. The ice-ocean turning angle is set to zero, reflecting that the water drag on the ice acts purely along the flow direction. Drag coefficient values related to wave characteristics are based on Smedsrud (2011) and Alberello et al. (2019). Finally, a range of wave parameters was selected to conduct a sensitivity analysis. These details will be added to the revised version of the manuscript.

**Referee #1** - L238: Why can we expect a higher growth rate for frazil ice? Is it because it is thinner than the floes?

Authors' response - The net surface heat flux, $F_0$, is thickness dependent: a smaller ice thickness results in a higher $F_0$, which in turn leads to a greater rate of change in thickness. Furthermore, the latent heat of fusion of frazil ice is lower than that of ice floes, also contributing to a higher rate of change in thickness. We will add these details in the revised manuscript.

**Referee #1** - Fig 2: Why is there not much happening in the first and last parts of the curves in Fig. 2a and the first part of Fig. 2b?

Authors' response - In Fig. 2a, the thickness of both sea ice and snow is zero during the summer months, i.e. no ice, as the air temperature exceeds the threshold for ice formation and growth. Ice begins to form and its thickness increases in winter, around June, when the air temperature drops. The sea ice thickness returns to zero in spring, i.e. sea ice is melting. Note that in the model, Fig. 2b,

snow growth commences only once the minimum snow threshold ($h_{s,\min} = 0.02\,\mathrm{m}$) has been exceeded.

**Referee #1** - Fig 2: What are the radiative and snow forcings used to obtain these plots? And what is used for the simulations presented in the results section?

Authors' response - The radiative and snow forcings used to produce the plots depend on the net surface heat flux, $F_0$, which is the resultant of all fluxes applied in the thermodynamic model (Hunke et al., 2015). These include the sensible heat flux, latent heat flux, incoming and outgoing longwave radiation, and incoming shortwave radiation. The thermodynamic model is forced by ERA5 as stated in the manuscript. In the revised manuscript, we will extend the description by adding the full list of atmospheric inputs used to derive the fluxes.The conditions used to produce Fig 2, which shows the output of the thermodynamic model run as stand-alone, are the same as those used for the simulations presented in the results section. This will be clarified in the revised manuscript.

**Referee #1** - L306-307: How was the thickness of ice determined in the SAR image?

Authors' response - The SAR image gives the intensity of the reflected radar signal, which can be assimilated to surface properties. This allowed us to distinguish different types of ice cover by assuming an intensity threshold to obtain an initial layout of the two different ice types: ice floes and grease ice. It is not possible to derive thickness information from this image, and we used reference visual observations from the SCALE-WIN22 research expedition (reference cited in the text). The thicknesses of the ice floes in the simulations were randomly assigned, starting from the minimum threshold of $0.1\,\mathrm{m}$ that was observed in the field for pancakes. The thickness assigned to grease ice was chosen to be slightly smaller than the smallest ice floe thickness. These details will be added to the revised version of the manuscript.

**Referee #1** - L307: The minimum threshold for what exactly?

Authors' response - Most sea ice models, e.g. Rousset et al. (2015), do not simulate the process of ice formation starting from frazil ice aggregation, e.g. Smedsrud (2011); Smedsrud and Martin (2015). New ice formation in open water is considered only when heat losses are equal to or greater than the enthalpy required to form this sea ice thickness. This will be clarified in the revised manuscript.

**Referee #1** - L312: If I understand correctly, the sub-domains are defined for diagnosis purpose only. Mentioning them here caused some confusion for me, as not much information was provided about what they represented. One solution could be to not mention them here, but describe them later in the context of Fig. 8.

Authors' response - We agree with the referee's suggestion and will move the introduction of the subdomains closer to Fig. 8 in the revised manuscript.

**Referee #1** - L317: 'that gradually increase in thickness'. This could be misinterpreted as an increase in time. I suggest rephrasing.

Authors' response - We will rephrase sentence L317 to read: 'Two circular floes of different sizes and thicknesses are linked by a narrow connection, with thickness spatially varying from one floe to the other.'

**Referee #1** - Fig 5: Could you please explain the motivation behind the configurations shown in panels b-e, including the wiggles in panel e?

Authors' response - In Fig. 5(b-e), the emphasis is on the orientation of the narrow connection relative to the imposed wave, which originates from the west. We found that the domain-averaged sea ice viscosity significantly depends on the connection's orientation (Fig. 14). Wiggles were included in Fig. 5e to demonstrate that the irregular shape induces more pronounced oscillations in the curves, as observed in Fig. 14d. This clarification will be added to the revised manuscript.

**Referee #1** - L338-339: 'The viscosity is locally affected by the propagating wave, since the rheology of the grease ice and ice floes is described as a function of thickness (Eq. (15) and (18)).' I don't understand how the first part of the sentence follows from the second part. Could you please explain in more detail?

Authors' response - We agree with the referee that this sentence is unclear. The intended meaning was that the viscosity of both ice floes and grease ice depends on thickness, which is modified by waves and consequently affects viscosity. However, this effect cannot be observed in Fig. 6; therefore, we have decided to remove the sentence from the revised manuscript.

**Referee #1** - L361-364: 'Figure 8(e) and (f) illustrate the relative difference in shear viscosity between the highest and lowest wave periods. The viscosity values in the case with waves from the west are significantly higher than those from the south, indicating a greater response to wave periods.' I don't understand how this indicates a greater response to wave periods. Could you clarify?

Authors' response - By 'greater response', we mean that the spatially averaged viscosity exhibits an increased sensitivity to wave periods when waves originate from the west, compared to waves coming from the south. In the revised manuscript, we will explicitly add the phrase 'increased sensitivity' to L361-364.

**Referee #1** - L369-370: The north-south orientation of what?

Authors' response - This refers to the north-south orientation of the incoming wave. We will add this clarification to the revised manuscript.

**Referee #1** - L370: Can this linear relationship be derived analytically? If so, this may be another useful output of the paper regarding parameterization of these effects in climate models.

Authors' response - We do not think such a relationship can be derived analytically through first principles, which was our driver for building this more complex numerical model. As stated in the manuscript, the linear relationship is an emerging empirical property from the numerical simulation results. This linear relationship between sea ice viscosity and ice floe percentage can be expressed analytically and used in parameterisations. Indeed, in the discussion (L522), we stated that the sea ice viscosity for 100% ice floes can be derived using Eq. 13, while the viscosity for 0% ice floes can be obtained from Eq. 18. Since the relationship with sea ice concentration is linear, these two data points (100% and 0% ice floes) are sufficient to define it.

**Referee #1** - L376-377: 'Based on the linear relationship presented in Fig. 9, we assume that the model resolves the smaller scales of the heterogeneous field, allowing us to extract properties at larger scales.' How does the linear relationship allow you to make this assumption?

Authors' response - Fig. 9 is based on the percentage of ice floes in each subdomain. Due to the strong scale invariance of mean sea ice viscosity from $840\,\mathrm{m} \times 840\,\mathrm{m}$ (smaller scale) to $5040\,\mathrm{m} \times 5040\,\mathrm{m}$ (larger scale), we made this assumption. We acknowledge that the sentence is unclear and convoluted, and more of a corollary of this section. We will first move the sentence L376-377 to L388, after discussing

the scaling analysis to better frame the discussion. Additionally, we will explicitly mention the strong scale invariance in this sentence, which would then read: 'Based on the inclusion of smaller scale processes that we assume realistic, the emergence of the linear relationship presented in Fig. 9 and the strong scale invariance of the mean viscosity of sea ice, we can use the model to extract properties at larger scales.'

**Referee #1** - L380-381: 'Therefore, we considered subdomain groups of increasing size from 1 to 6, with the latter corresponding to the full domain (see Fig. 8(a))'. This is not entirely clear to me. What are the sizes of these groups and how are they chosen spatially? What units does '1 to 6' have in the above sentence and what does it represent exactly? Perhaps a diagram would help?

Authors' response - The entire domain is partitioned into 36 subdomains each of dimension $840\,\mathrm{m} \times 840\,\mathrm{m}$, as shown in Fig. 8a. By subdomain groups of increasing size from 1 to 6, we refer to grouping the subdomains to form progressively larger ones. Ultimately, the full domain is covered by only one tile with size corresponding to $6 \times 6$ subdomains. We refer to grids of $1 \times 1 = 1$ subdomain (e.g. subdomain number 1), $2 \times 2 = 4$ subdomains (e.g. subdomain numbers $1-2$, $7-8$), $3 \times 3 = 9$ subdomains (e.g. subdomain numbers $1-3$, $7-9$, $13-15$), $4 \times 4 = 16$ subdomains (e.g. subdomain numbers $1-4$, $7-10$, $13-16$, $19-22$), $5 \times 5 = 25$ subdomains (e.g. subdomain numbers $1-5$, $7-11$, $13-17$, $19-23$, $25-29$) and $6 \times 6 = 36$ subdomains (subdomain numbers $1-36$). The use of 'e.g.' here is intentional because these represent only one of the possible combinations for each grid size of size $n \times n$ within the entire domain. The total number of possible combinations per grid size is listed in the second column of Table 3.

We acknowledge that L380-381 are not entirely clear. Therefore, in the revised manuscript, we will replace the term 'subdomain groups' with 'grid sizes', referring explicitly to sizes from $1 \times 1$ (i.e. $840\,\mathrm{m} \times 840\,\mathrm{m}$) to $6 \times 6$ (i.e. $5040\,\mathrm{m} \times 5040\,\mathrm{m}$). Additionally, we will modify and refer to Fig. 8a, to illustrate one example combination for each grid size.

**Referee #1** - L382: 'The results are presented in Table 3, showing a strong scale invariance from 800 m up to 5 km. The 800 m scale is already sufficient to capture the heterogeneity of the ice cover, and variations in ice type patterns do not affect the mechanical response at the larger scales up to 5 km.' Where do the numbers 500 m and 5 km come from? I am guessing they result from the size of each group, but this is hard to infer from just Table 3. Also, for clarity, please state explicitly what is invariant with scale.

Authors' response - Correct — to be precise, a grid size of $1 \times 1$ is equivalent to $840\,\mathrm{m} \times 840\,\mathrm{m}$, and a grid size of $6 \times 6$ is equivalent to $5040\,\mathrm{m} \times 5040\,\mathrm{m}$. The mean viscosity is invariant with scale. We will add this clarification to the revised manuscript.

**Referee #1** - L497: 'in the dynamic model by 3% in $t = 24\,\mathrm{h}$'. Should it be 'at $t = 24\,\mathrm{h}$'?

Authors' response - We agree with the referee that the 3% difference is reached only at $t = 24\,\mathrm{h}$. In the revised manuscript, we will update the text accordingly.

**Referee #1** - L507-508: 'degree of heterogeneity' is a bit ambiguous to me. I think ice floe percentage relative to grease ice would be clearer.

Authors' response - To avoid ambiguity, we will replace 'degree of heterogeneity' with 'ice floe percentage within each subdomain' in the revised manuscript.

**References**

Alberello, A., Onorato, M., Bennetts, L., Vichi, M., Eayrs, C., MacHutchon, K., and Toffoli, A.: Brief communication: Pancake ice floe size distribution during the winter expansion of the Antarctic marginal ice zone, The Cryosphere, 13, 41–48, https://doi.org/https://doi.org/10.5194/tc-13-41-2019, 2019.

Hibler III, W.: A dynamic thermodynamic sea ice model, Journal of physical oceanography, 9, 815–846, 1979.

Hunke, E. C., Lipscomb, W. H., Turner, A. K., Jeffery, N., and Elliott, S.: CICE: The Los Alamos Sea ice model documentation and software user's manual version 5.1 LA-CC-06-012, T-3 Fluid Dynamics Group, Los Alamos National Laboratory, 675, 15, 2015.

Leppäranta, M. and Hibler III, W.: The role of plastic ice interaction in marginal ice zone dynamics, Journal of Geophysical Research: Oceans, 90, 11 899–11 909, 1985.

Mehlmann, C. and Richter, T.: A modified global Newton solver for viscous-plastic sea ice models, Ocean Modelling, 116, 96–107, 2017.

Paul, F., Mielke, T., Schwarz, C., Schröder, J., Rampai, T., Skatulla, S., Audh, R. R., Hepworth, E., Vichi, M., and Lupascu, D. C.: Frazil Ice in the Antarctic Marginal Ice Zone, Journal of Marine Science and Engineering, 9, 647, 2021.

Rousset, C., Vancoppenolle, M., Madec, G., Fichefet, T., Flavoni, S., Barthélemy, A., Benshila, R., Chanut, J., Levy, C., Masson, S., and Vivier, F.: The Louvain-La-Neuve sea ice model LIM3.6: global and regional capabilities, Geoscientific Model Development, 8, 2991–3005, https://doi.org/10.5194/gmd-8-2991-2015, publisher: Copernicus GmbH, 2015.

Smedsrud, L. H.: Grease-ice thickness parameterization, Annals of Glaciology, 52, 77–82, 2011.

Smedsrud, L. H. and Martin, T.: Grease ice in basin-scale sea-ice ocean models, Annals of Glaciology, 56, 295–306, 2015.

---

## Author Comment (AC3)

**Comments from Referee #2, followed by the authors' responses**

**Referee #2** - In their manuscript 'WIce-FOAM 1.0: Coupled dynamic and thermodynamic modelling...', Rutger Marquart and coauthors describe an OpenFOAM-based sea ice model designed to study the effects of sea ice type ('ice floes' versus 'grease ice'), spatial orientation and arrangement of ice floes, and wave forcing (wave length and direction) on the net (domain-averaged) viscosity of the ice. The model considers two ice types with two different, prescribed rheology models, at ice concentration of 100%. The dynamic part of the model is coupled with a thermodynamic model, so that nonlinear coupling between ice dynamics and thermodynamics can be studied.

The manuscript is very well and clearly written, the figures illustrate well the topics discussed and the results obtained. The assumptions and limitations of the model and of the simulations are clearly stated. I find the results very interesting and relevant, as they might contribute to better understanding of sea ice rheology in the marginal ice zone, and thus to better parameterizations for sea ice models. In my opinion, after some rather minor corrections, the manuscript is worth publishing in GMD.

**Referee #2** - Line 90: 'As a result, interactions between ice floes are represented as continuous, churning contact of varying intensity rather than brief, forceful impacts. This justifies the exclusion of ice floe failure and fracture.'

It justifies the exclusion of failure and fracture due to floe-floe collisions, but not fracture in general, due to, e.g., ice convergence, shear deformation or bending of floes on waves. Maybe it's worth formulating more precisely?

Authors' response - We thank the referee for highlighting this important point. In the revised manuscript, we will clarify that our approach justifies the exclusion of failure and fracture due to floe-floe collision, but not fracture processes in general as pointed out by the referee.

We will revise the sentence accordingly to avoid confusion with other fracture mechanisms such as those driven by convergence, shear or wave-induced bending. The sentence in the updated manuscript will read: 'As a result, interactions between ice floes are represented as continuous, churning contact of varying intensity rather than brief, forceful impacts. This justifies the exclusion of ice floe failure and fracture due to floe-floe collisions. However, we acknowledge that other potential fracture mechanisms are also excluded, such as those driven by ice convergence, shear deformation, or wave-induced bending.'

**Referee #2** - Lines 101-103: I'd consider moving the references from line 101 to 103, just before equation (3). Equation (2) is quite obvious and doesn't require references.

Authors' response - We agree with the referee. In the revised manuscript, the references have been moved from line 101 to line 103, and the references for Equation (2) have been removed.

**Referee #2** - Line 138: 'apical plane'?

Authors' response - The apical plane refers to the upper surface of the ice exposed to the atmosphere (i.e., the ice–air interface). To improve clarity, we have clarified this in the text as 'upper (apical) surface'. We also recall that the basal plane refers to the lower surface of the ice, which is in contact with the ocean. We will clarify this in the revised manuscript.

**Referee #2** - Lines 193-194: 'For the ice floes, the one-dimensional thermodynamic model in the z-direction, developed by Tedesco et al. (2009), is applied to OpenFOAM cells associated with ice

floes to simulate thermodynamic variations in snow and ice thickness.' Does it mean that the ice floes in cells associated with grease ice do not grow thermodynamically? And vice versa, how is the growth of grease ice treated in cells classified as ice floes?

Earlier, line 87 states that 'cells are classified according to the predominant ice type - either ice floes or grease ice'. But what does it really mean? And, first of all, why is this classification necessary? Is the information on the surface area fraction covered with a given ice type not enough?

Authors' response - We apologise for the lack of clarity. Frazil/grease ice also grows thermodynamically, as shown in Figure 2b. In our model each computational cell is treated as being entirely occupied either by ice floes or by grease ice. The classification of a cell as an 'ice floe' or 'grease ice' cell is determined by the dominant ice type within that cell, based on the initial condition field derived from the SAR image. During the simulation, no transitions between ice types occur. We will modify the sentence to clarify that thermodynamics acts in both ice types.

We note that using the surface area fraction of each ice type as an alternative would not be compatible with the limitations of the Volume-of-Fluid (VoF) method employed. In the VoF framework, a scalar field $\alpha$ represents the ice type: $\alpha = 1$ corresponds to an ice floe, $\alpha = 0$ to grease ice, and intermediate values represent the interface, which does not have a direct physical meaning. Therefore, for both numerical stability and physical consistency, it is preferable to initialise each cell with a single ice type.

**Referee #2** - Line 254: 'we can interpolate the thermodynamic model'. Is 'interpolate' the right word here? What exactly is interpolated?

Authors' response - We agree with the referee that 'interpolate' is not the right word, as it could cause confusion. In the updated manuscript, we will remove this term and state that the thermodynamic model results at the hourly timescale are updated to a time step smaller than the hourly frequency of the forcing functions, which allows for the coupling between the two models. We will add this clarification to the revised manuscript.

**Referee #2** - Lines 338-339: 'The viscosity is locally affected by the propagating wave, since the rheology of the grease ice and ice floes is described as a function of thickness.' Is it really the thickness that's responsible for the observed differences in viscosity? And not the different form of the two rheology models? In other words, if the thickness of ice floes and grease ice was the same, would viscosity remain unaffected?

Authors' response - We agree with the referee that the form of the rheology is responsible for the observed differences in viscosity. If both materials had the same thickness, their viscosities would still differ because their rheologies are fundamentally different. What we aim to emphasize in lines 338-339 is that, within each rheological framework, viscosity also depends on the local ice thickness. The propagating wave modifies the ice thickness field, which in turn affects the viscosity according to the thickness-dependent terms in Eqs. (15) and (18). Hence, the local variations in viscosity shown in Fig. 6 are a combined effect of the wave-induced thickness changes and the distinct rheological behaviour of the two ice types.

We will add this clarification to the revised manuscript.

**Referee #2** - Line 369: 'the intercept at 0% represents the viscosity of grease ice' – which seems to be zero in Fig. 9. I think it requires a comment.

Authors' response - We agree with the referee that this requires an additional comment, as it may

appear that the viscosity at 0% ice floes is zero, which is not the case. The viscosity at 0% ice floes (100% grease ice) is approximately $440\,\mathrm{kg\,s^{-1}}$, as also indicated in the equations shown in the top-left corner of Fig. 9. This small value is not easily visible in the figure because the Y-axis is scaled in $10^8\,\mathrm{kg\,s^{-1}}$.

We will add this clarification to the revised manuscript.

**Referee #2** - Lines 369-370: 'The north-south orientation describes a linear relationship (see the equation in Fig. 9).' This sentence is unclear. Orientation of what? A relationship between what?

Authors' response - This refers to the north-south orientation of the incoming wave. The relationship is between sea ice viscosity and the percentage of ice floes in the domain. We will add this clarification to the revised manuscript.

**Referee #2** - Lines 376-377: 'Based on the linear relationship presented in Fig. 9, we assume that the model resolves the smaller scales of the heterogeneous field, allowing us to extract properties at larger scales.' Please explain why/how exactly the obtained relationship justifies this assumption.

Authors' response - We acknowledge that the sentence is unclear.

In the revised manuscript we will explicitly mention the strong scale invariance in this sentence as follows: 'Based on the inclusion of smaller scale processes that we assume realistic, the emergence of the linear relationship presented in Fig. 9 and the strong scale invariance of the mean viscosity of sea ice, we are confident that our results can be used to extract properties at larger scales as further discussed in Section 4.'

**Referee #2** - I find the results presented in Figs. 13 and 14 quite remarkable. The contribution of the bridge connecting the two floes to the total model surface area is very minor, well below 1%, presumably closer to 0.1% (as far as I can estimate it from the plots in Fig. 13), but its influence on the area-averaged viscosity, seen in Fig. 14, is at the level of 1%. To me, this suggests that the net viscosity is indeed very sensitive to the orientation of tiny (in terms of surface area fraction) 'ice elements', presumably much more so than Fig. 9 may suggest – as, in the 'real' case, different contributions from many elements with different orientations cancel out. (All this might be related to the discussion around line 510 in the manuscript?). This would mean that in situations with strong anisotropy of the ice cover very small changes e.g. in wave propagation direction may lead to significant changes in net viscosity.

Another interesting thing in Fig. 14 is that the panel (b) is qualitatively different from the other three: In this case, the viscosity increases with wave period – as it does in the cases shown in Fig. 7, but unlike in those in Fig. 14 a,c,d. What is the explanation for this behavior? I think this fact should be commented upon.

Authors' response - We are pleased that the referee appreciated the test cases and their results, which indeed required a large amount of work and interpretation. We fully agree with the referee on the role of geometry, which contributes to the novelty of the present manuscript. One of the key outcomes of this study is the influence of the orientation of narrow connections relative to the incoming wave. The domain-averaged sea ice viscosity is very sensitive to these narrow connections, even though they represent only a small fraction of the surface area. Demonstrating this effect was precisely the purpose of the test cases.

The main difference between Fig. 14(a) and Fig. 14(b), and the resulting order of the curves, arises

from the orientation of the bridge with respect to the incoming wave. When the bridge is perpendicular to the wave (Fig. 14(a)), both floes (including the bridge) behave more as a rigid body, making the system less sensitive to the wave period. In contrast, when the bridge is parallel to the wave direction (Fig. 14(b)), the floes can move more independently, creating a higher strain rate in the bridge and, consequently, a lower viscosity. Clearly, when the angle is changed to 45° (Fig. 14(c) and (d)), the floes and the bridge again move more as a rigid body rather than as two independent bodies, with the outcome being more similar to Fig. 14(a).

We will add this comment to the revised manuscript.

**Referee #2** - Lines 486-488: 'We also observed changes in the magnitude and oscillatory behaviour of the mean shear viscosity that are indicative of resonance at smaller scales that can be propagated to the kilometre scales.' Is there a possibility that the oscillations are related to the model setup (e.g., the periodic domain and the fact that the wavelength is adjusted to the domain size) and/or numerics (numerical schemes used etc.)?

Authors' response - We observe oscillations in most of the sea ice viscosity curves, which appear to be more pronounced at smaller wave periods (see Figs. A1 and A2). We would exclude issues with numerical convergence and stability of the scheme since these oscillations are independent on the time step. Our comment on the resonance was made to include a possible influence of the domain periodicity and wave lengths, and we will be more explicit about it in the revision. However, no oscillations are observed when the bridge is oriented perpendicular to the wave direction (Fig. 14(a)), suggesting that the orientation of the narrow connections also plays an important role in the resonance. We will add this consideration in the revised version.

**Referee #2** - Line 497: 'in t = 24h'. Meaning after 24h?

Authors' response - We agree with the referee. In the revised manuscript, we will update the text to 'at $t = 24\,\mathrm{h}$'.